# RTPrune: Reading-Twice Inspired Token Pruning
# for Efficient DeepSeek-OCR Inference

**Ben Wan** [1 2] **Yan Feng** [3 1] **Zihan Tang** [4] **Weizhe Huang** [1 3] **Yuting Zeng** [3 1] **Jia Wang** [2] **Tongxuan Liu** [1]

## Abstract

DeepSeek-OCR leverages visual–text compression to reduce long-text processing costs and accelerate inference, yet visual tokens remain prone to redundant textual and structural information. Moreover, current token pruning methods for conventional vision–language models (VLMs) fail to preserve textual fidelity due to improper compression mechanisms. By analyzing the decoding process of DeepSeek-OCR, we find that a distinct two-stage reading trajectory: the model initially prioritizes the majority of high-norm tokens, then subsequently redistributes its attention to the remaining ones. Motivated by this insight, we propose *RTPrune*, a two-stage token pruning method tailored for DeepSeek-OCR. In the first stage, we prioritize high-norm visual tokens that capture salient textual and structural information. In the second stage, the remaining tokens are paired and merged based on optimal transport theory to achieve efficient feature aggregation. We further introduce a dynamic pruning ratio that adapts to token similarity and textual density for OCR tasks, enabling a better efficiency–accuracy trade-off. Extensive experiments demonstrate state-of-the-art performance, as evidenced by 99.47% accuracy and 1.23× faster prefill on OmniDocBench, achieved with 84.25% token retention when applied to DeepSeek-OCR-Large. Code is released.

## 1. Introduction

Recent Vision-Language Models (VLMs) (Liu et al., 2023; 2024a;b; 2025) have achieved strong performance on multimodal understanding, particularly visual question answering (VQA) (Zhang et al., 2025a), yet they struggle with the

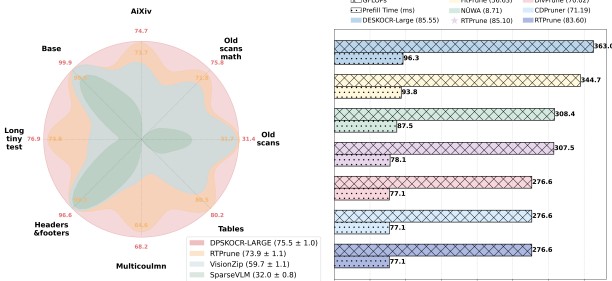

*Figure 1.* Performance and efficiency. (*left*) Our *RTPrune* consistently outperforms prior token pruning methods on DeepSeek-OCR, retaining over 97.88% of accuracy with 84% of visual tokens on olmOCR-Bench. (*right*) Our *RTPrune* reduces GFLOPs by nearly 15.29% and prefill time by nearly 18.90% on OmniDocBench when maintaining 99.47% accuracy.

fine-grained precision required for optical character recognition (OCR) tasks. DeepSeek-OCR (Wei et al., 2025) not only overcomes this challenge by substantially enhancing OCR capability, but also accelerates large language models (LLMs) inference by a small amount of visual tokens, demonstrating the effectiveness of visual modality as an efficient compression medium for textual information. However, despite being more compact than raw text, visual tokens remain prone to redundancy. In OCR scenarios, some visual tokens correspond to background regions or repeated structural patterns that contribute negligibly to text reconstruction, which indicates the potential to achieve further vision–text compression through visual token pruning.

Whereas, existing visual token pruning methods based on attention scores (Chen et al., 2024; Ye et al., 2025; Jiang et al., 2025a), textual relevance (Zhang et al., 2025b; 2024a), or inter-token similarities (Alvar et al., 2025; Wen et al., 2025; Yang et al., 2025) cannot be directly applicable to DeepSeek-OCR. First, unlike conventional VLMs, retraining the visual encoder in DeepSeek-OCR weakens the multimodal alignment between vision and language, leading to a misalignment between textual information and its original visual positions. Second, unlike query-driven VQA, OCR (Tang et al., 2026) aims to recover all textual content in an image, which severely constrains the aggressive compression while strictly preserving recognition accuracy. Therefore, DeepSeek-OCR calls for a dedicated token pruning method

---

[1]JD.com [2]Shanghai Jiao Tong University [3]University of Science and Technology of China [4]Tsinghua University. Correspondence to: Tongxuan Liu <liutongxuan1@jd.com>.

*Proceedings of the 43rd International Conference on Machine Learning*, Seoul, South Korea. PMLR 306, 2026. Copyright 2026 by the author(s).

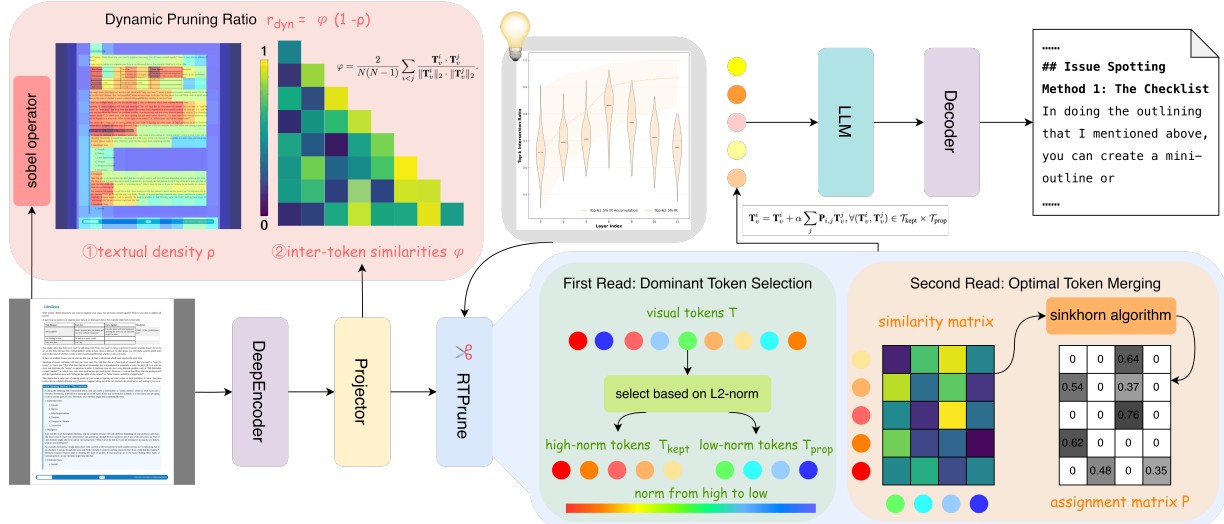

*Figure 2.* Overview of *RTPrune*. Our framework dynamically determines the pruning ratio by evaluating inter-token similarity and image-level textual density. The process then bifurcates into dominant token selection via embedding $\ell_2$-norms and residual information integration through optimal transport-based merging. As a training-free and model-agnostic approach, *RTPrune* mimics the dual-pass reading behavior of the LLM, preserving DeepSeek-OCR's performance while significantly reducing visual redundancy.

that not only retains text-informative visual tokens, but also automatically adapts the pruning ratio to each input image.

To address these issues, we leverage the fact that the vision encoder and LLM are jointly optimized in DeepSeek-OCR. Specially, we analyze attention variations across the LLM layers to investigate whether visual tokens receiving higher attention scores also exhibit larger $\ell_2$-norms post-encoding. The results reveal a distinct two-stage reading behavior within the model. While the model predominantly attends to the majority of high-norm visual tokens in the shallow layers due to their salient textual and structural information, attention progressively shifts toward other tokens, including the remaining high-norm ones, in the deeper layers to incorporate complementary cues. As such, the model focuses on high-norm tokens and effectively aggregates remaining tokens across the LLM layers.

Motivated by this observation, we propose *RTPrune*, a plug-and-play inference acceleration method for DeepSeek-OCR. In the first stage, *RTPrune* prioritizes visual tokens with larger feature $\ell_2$-norms, which capture the most salient textual and structural information, emulating the LLM's attention focus. In the second stage, to reflect the attention redistribution observed in deeper layers, the remaining tokens are matched via optimal transport and adaptively merged into selected tokens, allowing complementary structural and contextual cues to be preserved despite token reduction. Furthermore, *RTPrune* adopts a dynamic pruning ratio to meet the high-precision requirements of OCR tasks. By jointly evaluating token redundancy through feature similarity and textual density via edge-intensity detection, this strategy effectively discards non-textual regions while retaining essential informative cues, enabling *RTPrune* a better efficiency–accuracy trade-off. As shown in Figure 1, when applied in DeepSeek-OCR-Large, our *RTPrune* achieving 99.47% accuracy and 1.23× faster prefill on OmniDocBench (Ouyang et al., 2025) with only 84.25% token retention. In summary, our work makes three principal contributions:

- We introduce *RTPrune*, a plug-and-play visual token pruning method in DeepSeek-OCR which mimics the reading twice behavior of the LLM via a two-stage pipeline: retaining high-norm tokens and then merging the remaining ones via optimal transport.

- We propose a dynamic pruning strategy to enable a better efficiency–accuracy trade-off, which combines the post-encoding inter-token similarity and the original textual density of the image.

- We conduct extensive experiments on various OCR benchmarks, demonstrating that *RTPrune* consistently achieves state-of-the-art under both a fixed pruning ratio and our dynamic pruning strategy.

## 2. Related Work

**DeepSeek-OCR.** Unlike general VLMs (Liu et al., 2023; 2024a;b; Zhang et al., 2024b; Bai et al., 2025) focusing on multimodal understanding, DeepSeek-OCR takes visual-text compression as the core goal, leveraging visual modality to efficiently compress textual information for the LLM long-context processing. Different from previous OCR models (Cui et al., 2025; Poznanski et al., 2025a; Li et al.,

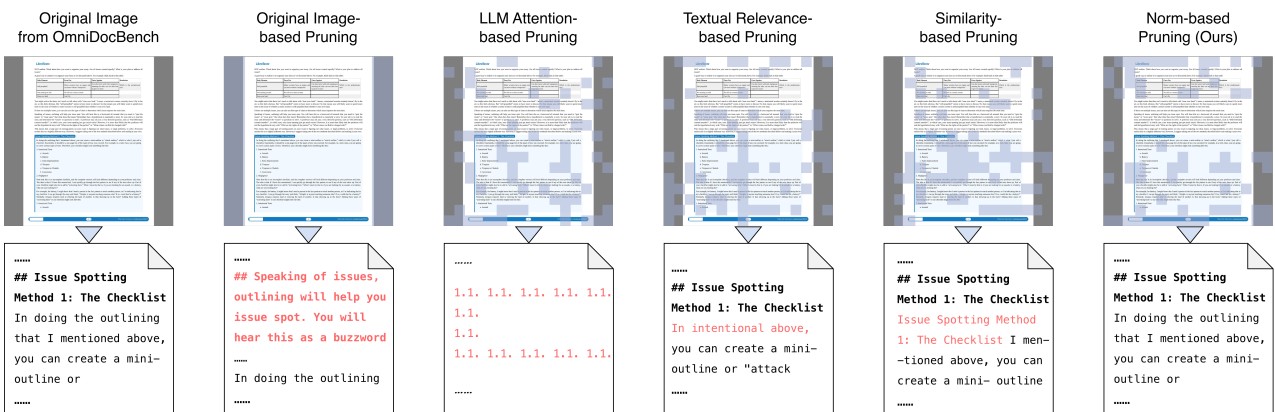

*Figure 3.* Comparison of different token pruning methods on DeepSeek-OCR-Base. The patches highlighted in blue are pruned and the text highlighted in red indicates discrepancies with the ground truth. While methods based on original image, attention scores, textual relevance, or inter-token similarity fail to generate text accurately, our approach precisely captures tokens containing critical textual information, leading to superior OCR performance.

*Table 1.* Performance comparison of various pruning methods on OmniDocBench (10% subset) using DeepSeek-OCR-Base at a 25% pruning. Text and Read Order are measured by *Edit Distance↓*; Formula is measured by *CDM (Character Distribution Match)↑*; Table is measured by *TEDS (Tree Edit Distance-based Similarity)↑* and *TEDS-S↑*.

| METHODS | TEXT | FORMULA | TABLE | ORDER | OVERALL |
|---|---|---|---|---|---|
| BASELINE | 0.18 | 83.47 | 88.82 / 92.74 | 0.16 | 84.86 |
| RANDOM | 0.52 | 61.72 | 62.02 / 72.80 | 0.30 | 57.24 |
| FASTV(ECCV24) | 0.90 | 5.02 | -0.29 / 2.27 | 0.63 | 4.78 |
| DIVPRUNE(CVPR25) | 0.36 | 70.67 | 61.14 / 72.33 | 0.25 | 65.30 |
| CDPRUNER(NIPS25) | 0.47 | 55.74 | 47.39 / 60.92 | 0.31 | 52.01 |

2025b), it adopts a custom DeepEncoder and MoE decoder, achieving up to 20× compression ratio with far fewer vision tokens while supporting complex content parsing, integrating practical OCR capabilities with long-context compression research value.

**Visual token pruning.** Visual token pruning (Deng et al., 2025; Li et al., 2025a; Jiang et al., 2025b; Zhang et al., 2025b; Tong et al., 2026; Zou et al., 2026; Yao et al., 2026) aims to reduce the number of visual tokens processed by VLMs while preserving task performance, thereby improving inference efficiency. Existing methods primarily estimate the importance or redundancy of visual tokens and selectively retain a compact subset for downstream processing. According to the criteria used for token selection, prior work can be broadly categorized into several types. Attention-based methods leverage attention scores from the vision encoder or the LLM to identify salient tokens, such as FastV (Chen et al., 2024), SparseVLM (Zhang et al., 2024a), and FitPrune (Ye et al., 2025). Similarity-based methods focus on removing redundant tokens by measuring inter-token similarity or diversity, exemplified by DivPrune (Alvar et al., 2025) and DART (Wen et al., 2025). Hybrid attention- and similarity-based approaches combine both signals to achieve more robust pruning, including VisionZip (Yang et al., 2025) and NÜWA (Huang et al., 2026). In addition, textual relevance–based methods incorporate similarity between visual tokens and textual prompts to guide pruning decisions, as in CDPruner (Zhang et al., 2025b). While effective for VQA-centric VLMs, these methods rely on assumptions about semantic alignment or query relevance that do not directly extend to OCR-oriented models such as DeepSeek-OCR.

## 3. Method

In this section, we first conduct a pilot study to analyze why existing token pruning methods fail when applied to DeepSeek-OCR in Section 3.1. We then explore the decodeing process of the LLM to identify opportunities for effective compression in Section 3.2. Finally, we present our *RTPrune* in Section 3.3 to satisfy the high-precision requirement of OCR task. The overall design of *RTPrune* is shown in Figure 2.

### 3.1. Limitations of Existing Methods

To evaluate the performance of token pruning methods—originally designed for general VLMs (Liu et al., 2023)—on DeepSeek-OCR, we conduct a pilot study on a 10% subset of the OmniDocBench dataset (Ouyang et al., 2025). Table 1 reports the performance of various pruning methods on DeepSeek-OCR-Base at a 25% pruning ratio, while Figure 3 provides a visual comparison of representative results. It is evident that pruning methods designed for conventional VLMs perform even worse than random pruning when applied to DeepSeek-OCR. The generated outputs often suffer from omissions, repetitive patterns, or the generation of nonsensical content.

This phenomenon can be attributed to fundamental differences in both task characteristics and model architectures. From a task perspective, unlike VQA which targets sparse, question-relevant regions, OCR requires exhaustive, high-fidelity reconstruction of all textual and layout details. Consequently, pruning strategies based on token diversity (Alvar et al., 2025) are ill-suited for the dense nature of OCR. Furthermore, methods relying on prompt-text relevance (Zhang et al., 2025b) fail, as the uniformity of prompts in OCR provides insufficient semantic guidance to distinguish important regions containing critical textual information. Besides, from a model perspective, while conventional VLMs use frozen, pre-aligned encoders (Radford et al., 2021; Zhai et al., 2023) with stable semantic-visual coupling, DeepSeek-OCR retrains its encoder for dense reconstruction. This process significantly weakens explicit vision–language alignment, causing textual information to be spatially distributed and reducing the interpretability of encoder-side attention signals. Furthermore, the extreme information density of OCR inputs causes different LLM layers to attend to distinct subsets of visual tokens. This layer-wise divergence renders pruning strategies based on LLM-side attention (Chen et al., 2024; Zhang et al., 2024a; Ye et al., 2025) largely ineffective. Detailed discussions are in Section A.

### 3.2. Attention Vary Observation in LLM Decoding

For DeepSeek-OCR, we must fundamentally reconsider where textual information actually resides within the visual space and how informative tokens can be identified. This reconsideration is closely tied to the model design of DeepSeek-OCR. From a model perspective, it employs a strategy of retraining the vision encoder (Wei et al., 2024) alongside the LLM when prefill, rather than keeping the vision backbone frozen. This raises a natural question: do visual tokens that receive higher attention in the LLM also tend to be assigned larger $\ell_2$-norms by the encoder (Darcet et al., 2024; Lappe & Giese, 2025)? To investigate this, we select Top-K visual tokens by their embedding $\ell_2$-norms and evaluate the Intersection Ratio (TIR) with Top-K high-attention tokens at each LLM layer, as well as with the union of such tokens from the current and all previous layers, as shown in Figure 4.

We observe a strong correlation between embedding $\ell_2$-norms and LLM attention. The layer-wise TIR increases and then decreases across layers, whereas the cumulative TIR increases monotonically, which rises rapidly in shallow layers and more gradually in deeper ones. Notably, even in the Top-3/8 setting, the maximum intersection ratio reaches 64.4%, while the final cumulative intersection ratio attains 88.71%. This pattern indicates a reading twice behavior during OCR generation: shallow layers focus on a subset of high-norm tokens that encode rich textual and structural

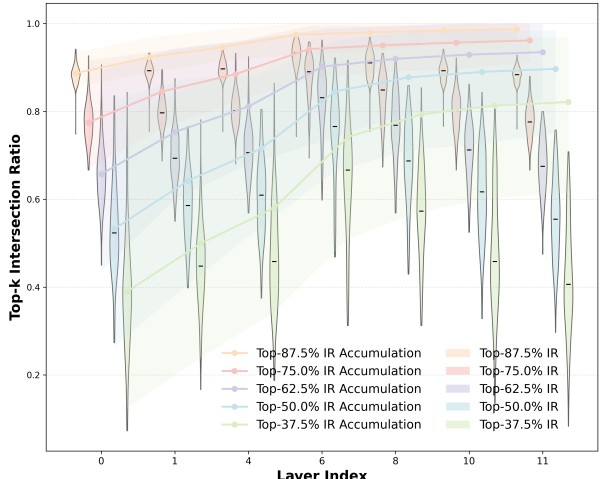

*Figure 4.* Top-K Intersection Ratio (TIR) between high-norm visual embeddings and LLM high-attention viusal tokens during prefill, where the violin plot shows the TIR with high-attention tokens at each individual LLM layer, while the line plot reports the TIR with the union of high-attention tokens from the current and all previous layers, evaluated on OmniDocBench (10% subset) by DeepSeek-OCR-Base.

information, while deeper layers attend to additional tokens, including the remaining high-norm ones, to implement content and structure. As such, the model effectively traverses high-norm tokens and integrates remainings across the LLM layers. Motivated by this observation, we propose a two-stage token pruning approach, termed *RTPrune*. In the first stage, tokens are pruned according to their embedding $\ell_2$-norms; in the second stage, the retained tokens absorb information from the discarded ones via merging.

### 3.3. RTPrune

#### 3.3.1. DOMINANT TOKEN SELECTION

DeepSeek-OCR consists of three core components: a vision encoder $f_v$, a multimodal projector $g$, and a large language model $f_\phi$. The vision encoder encodes the input image $\mathbf{I}$ into a sequence of visual tokens $\mathbf{T}_v = g(f_v(\mathbf{I})) \in \mathbb{R}^{N \times D}$, where $N$ is the number of visual tokens and $D$ is the dimension of visual tokens. Drawing upon our prior findings, the LLM exhibits a heightened focus on visual tokens with larger embedding $\ell_2$-norms during the initial reading pass, as these tokens are likely to contain dense textual information. Consequently, we utilize the embedding $\ell_2$-norm as the primary criterion for our dominant token selection strategy:

$$\mathcal{C}_k = \|\mathbf{T}_v^k\|_2. \tag{1}$$

Suppose the target pruning ratio is $r$, and the number of tokens to be retained is $M = N(1-r)$. We define the selected token set $\mathcal{T}_{\text{kept}}$ as the top-$M$ tokens ranked by Equation (1), while the remaining tokens constitute the proposed-to-prune

set $\mathcal{T}_{\text{prop}}$.

### 3.3.2. OPTIMAL TOKEN MERGING

As previously noted, DeepSeek-OCR exhibits a distinctive attention pattern where the LLM first focuses on large-norm visual tokens and subsequently shifts focus. Accordingly, we design a two-stage strategy: the first stage identifies information-rich tokens via norm-based selection, while the second stage employs an optimal transport-based (Villani et al., 2008; Cuturi, 2013) token merging mechanism to recover essential cues and compensate for the information loss incurred during pruning.

The token merging process is implemented via an assignment matrix $\mathbf{P} \in \mathbb{R}^{M \times (N-M)}$, which can be obtained by following optimal transport problem:

$$\max \quad \sum_{i,j} \mathbf{S}_{i,j} \cdot \mathbf{P}_{i,j} \tag{2}$$
$$\text{s.t.} \quad \mathbf{P}\mathbf{1}_{N-M} = \mathbf{1}_M \quad \text{and} \quad \mathbf{P}^{\mathrm{T}}\mathbf{1}_M = \mathbf{1}_{N-M},$$

where $\mathbf{S} \in \mathbb{R}^{M \times (N-M)}$ is a score matrix, $\mathbf{1}$ denotes the all-ones vector and our goal is maximizing the total score for all possible matches. We express the pairwise score as the similarity of matching visual tokens:

$$\mathbf{S}_{i,j} = \frac{\mathbf{T}_v^i \cdot \mathbf{T}_v^j}{\|\mathbf{T}_v^i\|_2 \cdot \|\mathbf{T}_v^j\|_2}, \forall (\mathbf{T}_v^i, \mathbf{T}_v^j) \in \mathcal{T}_{\text{kept}} \times \mathcal{T}_{\text{prop}}. \tag{3}$$

Considering that certain tokens exhibit significant discrepancies from others, merging them might compromise the integrity of the information carried by the original tokens. Therefore, we augment the scores $\mathbf{S}$ to $\bar{\mathbf{S}} \in \mathbb{R}^{(M+1) \times (N-M+1)}$ by appending a new row and column (Sarlin et al., 2020), filled with a fixed parameter $z$:

$$\bar{\mathbf{S}}_{i,N-M+1} = \bar{\mathbf{S}}_{M+1,j} = \bar{\mathbf{S}}_{M+1,N-M+1} = z \in \mathbb{R}. \tag{4}$$

While proposed-to-prune tokens in $\mathcal{T}_{\text{prop}}$ will be merged to some selected tokens in $\mathcal{T}_{\text{kept}}$ or the dustbin which has as many matches as there are tokens in the other set. The augmented assignment $\bar{\mathbf{P}}$ now has new constraints in Equation (2):

$$\bar{\mathbf{P}}\mathbf{1}_{N-M+1} = \boldsymbol{a} \quad \text{and} \quad \bar{\mathbf{P}}^{\mathrm{T}}\mathbf{1}_{M+1} = \boldsymbol{b}, \tag{5}$$

where $\boldsymbol{a} = [\mathbf{1}_M^{\mathrm{T}} \quad N-M]^{\mathrm{T}}$ and $\boldsymbol{b} = [\mathbf{1}_{N-M}^{\mathrm{T}} \quad M]^{\mathrm{T}}$ are expected matches for each token and dustbin in $\mathcal{T}_{\text{kept}}$ and $\mathcal{T}_{\text{prop}}$.

Above optimal transport problem can be efficiently solved by Sinkhorn algorithm (Cuturi, 2013) and the solution can be obtained by dropping the dustbins $\mathbf{P} = \bar{\mathbf{P}}_{:-1,:-1}$ after several iterations. The algorithm implementation is provided in Section C. The selected tokens are merged with the

proposed-to-prune by $\mathbf{P}$:

$$\mathbf{T}_v^i = \mathbf{T}_v^i + \alpha \sum_j \mathbf{P}_{i,j} \cdot \mathbf{T}_v^j, \forall (\mathbf{T}_v^i, \mathbf{T}_v^j) \in \mathcal{T}_{\text{kept}} \times \mathcal{T}_{\text{prop}}, \tag{6}$$

where $\alpha$ is a hyperparameter to control the merge strength.

### 3.3.3. DYNAMIC PRUNING RATIO

While VLMs (Liu et al., 2023; 2024a;b; Zhang et al., 2024b; Bai et al., 2025) in VQA tasks can sustain high compression ratios without significant performance degradation, OCR demands a much denser token set to achieve high-fidelity output. This challenge is further amplified in DeepSeek-OCR, which is already optimized for visual efficiency.

Although encoded visual tokens encapsulate critical textual information, they inevitably incorporate semi-essential elements such as background and layout information. To achieve a better efficiency–accuracy trade-off, it is imperative to prune these redundant, non-textual regions without disturbing informative tokens. To this end, we introduce a dynamic pruning strategy guided by two complementary indicators: inter-token similarity and textual density.

**Inter-token similarity.** The non-textual components mentioned above often exhibit high feature-level correlation, leading to representation redundancy. To quantify this global feature homogeneity, we introduce the average inter-token similarity $\varphi$:

$$\varphi = \frac{2}{N(N-1)} \sum_{i<j} \frac{\mathbf{T}_v^i \cdot \mathbf{T}_v^j}{\|\mathbf{T}_v^i\|_2 \cdot \|\mathbf{T}_v^j\|_2}. \tag{7}$$

A higher $\varphi$ suggests a more uniform feature space, indicating that the visual content is structurally repetitive and thus possesses a higher theoretical capacity for reduction.

**Textual density.** However, high similarity alone does not guarantee that a region is expendable, as dense text can also exhibit correlation. To ensure that pruning targets only non-textual information, we define the textual density $\rho$ as a safeguard to estimate the information richness of the document image. Textual regions in document images are characterized by high-frequency edges and sharp intensity transitions, which can be effectively captured using the Sobel operator (Sobel et al., 1968), whose implementation is provided in Section B. Specifically, for each token patch $\mathbf{I}_k, k = 1, \ldots, N$ containing $h \times w$ pixels, we first calculate the gradient magnitude $G(i,j) = \sqrt{G_x(i,j)^2 + G_y(i,j)^2}$ for each pixel $(i,j) \in \mathbf{I}_k$. A pixel is identified as an active edge pixel if its gradient magnitude exceeds a predefined threshold $\tau$. The local density $\rho_k$ for a single patch is thus defined as the ratio of active edge pixels to the total number of pixels within the patch:

$$\rho_k = \frac{1}{h \times w} \sum_{i,j}^{h,w} \mathbb{I}(G(i,j) \geq \tau), \tag{8}$$

*Table 2.* Performance comparison of various pruning methods on OmniDocBench using DeepSeek-OCR. Methods only pruning at vision encoder are shown with green background, methods only pruning within LLM with orange background, multi-stage pruning methods with blue background and our method with pink background. The best-performing result in each column is **bolded**.

| METHOD | DYNAMIC | #TOKENS | TEXT$^{\text{EDIT}}$ ↓ | FORMULA$^{\text{CDM}}$ ↑ | TABLE$^{\text{TEDS}}$ ↑ | TABLE$^{\text{TEDS-S}}$ ↑ | ORDER$^{\text{EDIT}}$ ↓ | OVERALL ↑ |
|---|---|---|---|---|---|---|---|---|
| | | | | DEEPSEEK-OCR-TINY | | | | |
| BASELINE | - | 64 | 0.27 | 70.01 | 62.86 | 72.97 | 0.20 | 68.59 |
| FASTV(ECCV24) | ✗ | 48 | 0.95 | 2.20 | 0.26 | 1.17 | 0.68 | 2.49 |
| DART(EMNLP25) | ✗ | 48 | 0.96 | 1.39 | -0.10 | 0.34 | 0.72 | 1.73 |
| *RTPrune* | ✗ | 48 | 0.48 | 50.38 | 40.22 | 52.77 | 0.42 | 47.60 |
| SPARSEVLM(ICML25) | ✔ | AVG=52 | 0.89 | 6.31 | 5.07 | 9.92 | 0.64 | 7.50 |
| VISIONZIP(CVPR25) | ✔ | AVG=52 | 0.50 | 50.37 | 41.44 | 56.06 | **0.33** | 47.37 |
| *RTPrune* | ✔ | AVG=52 | **0.43** | **53.98** | **45.68** | **57.56** | 0.38 | **52.09** |
| | | | | DEEPSEEK-OCR-SMALL | | | | |
| BASELINE | - | 100 | 0.17 | 79.14 | 76.58 | 82.43 | 0.15 | 79.61 |
| FITPRUNE(AAAI25) | ✗ | 75 | 0.78 | 17.63 | 13.13 | 17.93 | 0.61 | 17.55 |
| NÜWA(ICLR26) | ✗ | 75 | 0.92 | 4.13 | 2.38 | 5.97 | 0.68 | 5.00 |
| *RTPrune* | ✗ | 75 | 0.37 | 60.74 | 53.71 | 64.35 | 0.33 | 59.08 |
| DIVPRUNE(CVPR25) | ✔ | AVG=83 | 0.39 | 59.90 | 55.48 | 68.36 | **0.26** | 58.93 |
| CDPRUNER(NIPS25) | ✔ | AVG=83 | 0.45 | 63.46 | **60.61** | **70.05** | 0.30 | 59.75 |
| *RTPrune* | ✔ | AVG=83 | **0.31** | **66.32** | 58.90 | 67.86 | 0.28 | **64.91** |
| | | | | DEEPSEEK-OCR-BASE | | | | |
| BASELINE | - | 256 | 0.12 | 81.90 | 84.58 | 88.43 | 0.11 | 84.83 |
| SPARSEVLM(ICML25) | ✗ | 192 | 0.74 | 27.39 | 21.07 | 30.09 | 0.50 | 24.92 |
| VISIONZIP(CVPR25) | ✗ | 192 | 0.37 | 67.09 | 59.39 | 72.31 | 0.23 | 63.29 |
| *RTPrune* | ✗ | 192 | 0.21 | 77.27 | 75.55 | 81.88 | 0.19 | 77.37 |
| FASTV(ECCV24) | ✔ | AVG=214 | 0.87 | 7.42 | 3.95 | 5.61 | 0.65 | 8.12 |
| DART(EMNLP25) | ✔ | AVG=214 | 0.58 | 33.95 | 33.37 | 38.54 | 0.46 | 36.47 |
| *RTPrune* | ✔ | AVG=214 | **0.17** | **79.95** | **81.60** | **86.74** | **0.16** | **81.48** |
| | | | | DEEPSEEK-OCR-LARGE | | | | |
| BASELINE | - | 400 | 0.09 | 81.70 | 83.87 | 88.03 | 0.09 | 85.55 |
| DIVPRUNE(CVPR25) | ✗ | 300 | 0.26 | 71.22 | 64.83 | 75.39 | 0.17 | 70.02 |
| CDPRUNER(NIPS25) | ✗ | 300 | 0.32 | 73.60 | 72.16 | 80.38 | 0.20 | 71.19 |
| *RTPrune* | ✗ | 300 | 0.13 | 81.71 | 81.79 | 85.90 | 0.13 | 83.60 |
| FITPRUNE(AAAI25) | ✔ | AVG=337 | 0.30 | 55.61 | 44.17 | 46.97 | 0.24 | 56.63 |
| NÜWA(ICLR26) | ✔ | AVG=337 | 0.85 | 5.50 | 6.04 | 11.73 | 0.64 | 8.71 |
| *RTPrune* | ✔ | AVG=337 | **0.10** | **82.70** | **82.11** | **86.51** | **0.09** | **85.10** |

where $\mathbb{I}(\cdot)$ is the indicator function. The global textual density $\rho = \text{avg}(\rho_k)$ for the entire image is then computed as the average density across all patches. a lower $\rho$ indicates that the image is dominated by margins or backgrounds, which are primary candidates for pruning.

**Dynamic pruning ratio.** By synthesizing these two perspectives, we define a dynamic pruning ratio $r_{\text{dyn}}$ that scales according to the identified non-textual redundancy:

$$r_{\text{dyn}} = f_{\text{normalize}}(\varphi)(1 - \rho). \qquad (9)$$

By scaling with feature overlap and inversely with textual density, this ratio effectively targets non-textual regions while safeguarding critical informative cues. This mechanism facilitates content-aware pruning, which achieves a more optimal balance between efficiency and accuracy.

## 4. Experiments

### 4.1. Experimental Setup

**Evaluation benchmarks.** To thoroughly assess the effectiveness of *RTPrune* on DeepSeek-OCR, we evaluate our method on three OCR-based benchmarks, including OmniDocBench (Ouyang et al., 2025), olmOCR-Bench (Poznanski et al., 2025b) and Ocean-OCR benchmark (Chen et al., 2025). All experiments follow the default settings and evaluation metrics of these benchmarks. Detailed descriptions are provided in the Section E.

**Comparison methods.** We choose several recent works of different types as comparison methods, including methods only pruning at vision encoder like VisionZip (Yang et al., 2025), DivPrune (Alvar et al., 2025) and CDPruner (Zhang et al., 2025b), methods only pruning within LLM like FastV (Chen et al., 2024), DART (Wen et al., 2025) and FitPrune (Ye et al., 2025), as well as multi-stage pruning methods like SparseVLM (Zhang et al., 2024a) and NÜWA (Huang et al., 2026). We evaluate each method at least once under the same settings to ensure a comprehensive comparison.

### 4.2. Main Results

**OmniDocBench.** We first evaluate *RTPrune* on OmniDocBench, a widely adopted benchmark for assessing diverse document parsing tasks in real-world scenarios. Ta-

*Table 3.* Performance comparison of various pruning methods on olmOCR-Bench using DeepSeek-OCR. Methods only pruning at vision encoder are shown with green background , methods only pruning within LLM with orange background , multi-stage pruning methods with blue background and our method with pink background . The best-performing result in each column is **bolded**.

| METHOD | DYNAMIC | #TOKENS | AIXIV | OLD SCANS MATH | TABLES | OLD SCANS | HEADERS & FOOTERS | MULTI COLUMN | LONG TINY TEXT | BASE | OVERALL |
|---|---|---|---|---|---|---|---|---|---|---|---|
| DEEPSEEK-OCR-BASE | | | | | | | | | | | |
| BASELINE | - | 256 | 69.1 | 76.6 | 77.7 | 30.0 | 95.8 | 65.3 | 65.6 | 99.3 | 72.4 ± 1.0 |
| FITPRUNE(AAAI25) | ✗ | 192 | 12.9 | 25.8 | 27.2 | 17.1 | 92.6 | 5.1 | 4.8 | 87.6 | 34.1 ± 0.8 |
| NÜWA(ICLR26) | ✗ | 192 | 0.2 | 2.2 | 2.7 | 14.1 | **97.1** | 0.0 | 0.0 | 70.7 | 23.4 ± 0.6 |
| *RTPrune* | ✗ | 192 | 59.5 | 65.3 | 62.8 | 28.5 | 94.3 | 31.8 | 20.4 | 98.6 | 57.7 ± 1.0 |
| DIVPRUNE(CVPR25) | ✔ | AVG=213 | 42.2 | 42.8 | 48.0 | 25.1 | 95.7 | 22.3 | 25.8 | 98.9 | 50.1 ± 1.1 |
| CDPRUNER(NIPS25) | ✔ | AVG=213 | 47.2 | 56.5 | 65.5 | 24.5 | 95.1 | 15.0 | 21.5 | 98.9 | 53.0 ± 1.1 |
| *RTPrune* | ✔ | AVG=213 | **65.1** | **69.7** | 71.1 | **29.5** | 95.3 | **45.1** | **34.8** | **99.1** | **63.7 ± 1.1** |
| DEEPSEEK-OCR-LARGE | | | | | | | | | | | |
| BASELINE | - | 400 | 74.7 | 75.8 | 80.2 | 31.4 | 96.6 | 68.2 | 76.9 | 99.9 | 75.5 ± 1.0 |
| FASTV(ECCV24) | ✗ | 300 | 1.4 | 3.1 | 2.8 | 15.0 | 97.1 | 0.1 | 0.2 | 82.1 | 25.2 ± 0.5 |
| DART(EMNLP25) | ✗ | 300 | 1.2 | 3.4 | 3.0 | 15.4 | **98.6** | 0.1 | 0.1 | 84.5 | 25.6 ± 0.5 |
| *RTPrune* | ✗ | 300 | 72.8 | 68.6 | 78.2 | 31.0 | 95.8 | 59.0 | 42.1 | 99.0 | 68.3 ± 1.1 |
| SPARSEVLM(ICML25) | ✔ | AVG=336 | 16.1 | 6.3 | 12.5 | 16.3 | 93.6 | 6.2 | 17.6 | 87.3 | 32.0 ± 0.8 |
| VISIONZIP(CVPR25) | ✔ | AVG=336 | 58.0 | 60.9 | 63.4 | 30.8 | 96.4 | 27.7 | 41.2 | 99.0 | 59.7 ± 1.1 |
| *RTPrune* | ✔ | AVG=336 | **73.7** | **71.8** | **80.5** | **31.7** | 95.3 | **64.6** | **73.8** | **99.5** | **73.9 ± 1.1** |

ble 2 reports the performance of different pruning methods across multiple DeepSeek-OCR variants under both a fixed 25% pruning ratio and our proposed dynamic pruning strategy. Under both settings, *RTPrune* consistently achieves the best performance among all compared approaches. Notably, with 15.75% of tokens pruned, *RTPrune* remarkably maintains 99.47% the original performance of DeepSeek-OCR-Large. This advantage arises from our analysis of the LLM decoding process, which reveals that the embedding norm in DeepSeek-OCR serves as an effective indicator of textual information content. In contrast, methods that perform token pruning within the LLM, such as FitPrune (Ye et al., 2025) and SparseVLM (Zhang et al., 2024a), typically prune tokens at shallow-to-middle layers to maximize inference speed, thereby preventing the model from fully exploiting critical information emphasized in later layers and resulting in notable performance degradation. Meanwhile, approaches that apply token pruning only at the vision encoder stage, such as VisionZip (Yang et al., 2025) and CDPruner (Zhang et al., 2025b), rely on encoder attention or post-encoding feature diversity and thus fail to accurately capture complete textual information, inevitably leading to degraded OCR accuracy.

**olmOCR-Bench.** We further evaluate *RTPrune* on olmOCR-Bench, which provides a rigorous, unit-test-based evaluation framework for PDF content extraction. Table 3 summarizes the performance of different pruning methods across multiple DeepSeek-OCR variants under both a fixed 25% pruning ratio and our dynamic pruning strategy. Existing approaches, as well as *RTPrune* with a fixed 25% pruning ratio, exhibit substantial performance degradation on challenging subsets such as multicolumn and long tiny

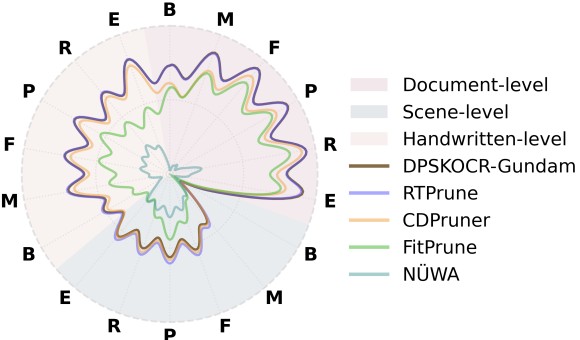

*Figure 5.* Performance comparison of various pruning methods with dynamic pruning strategy on ocean-OCR Benchmark using DeepSeek-OCR-Gundam across multiple noteworthy OCR abilities. *E, F, P, R, B,* and *M* are the abbreviations for *Edit Distance, F1-Score, Precision, Recall, BLEU, and METEOR* respectively. For *Edit Distance*, the plotted score is computed with $x_{after} = 1 - x_{before}$ for better visualization.

text. This degradation arises not only from the inability of some methods to accurately identify visual tokens corresponding to textual content, but also from inappropriate pruning ratios that remove critical structural information. In contrast, *RTPrune* with the dynamic pruning strategy consistently delivers the best performance across all compared methods, retaining up to 97.88% accuracy with 84% of visual tokens. This advantage stems from its ability to capture the LLM's two-stage reading behavior while adaptively determining suitable pruning ratios that eliminate redundant, non-textual regions without disrupting informative tokens.

**Ocean-OCR benchmark.** To evaluate our method on the

*Table 4.* Efficiency analysis of different pruning methods on olmOCR-Bench using DeepSeek-OCR. Methods only pruning at vision encoder are shown with green background , methods only pruning within LLM with orange background , multi-stage pruning methods with blue background and our method with pink background .

| METHOD | DYNAMIC | #VISUAL TOKENS | #TOTAL TOKENS | GFLOPS | PREFILL TIME | DECODE TIME | PERFORMANCE |
|---|---|---|---|---|---|---|---|
| | | | DEEPSEEK-OCR-BASE | | | | |
| BASELINE | - | 256 | 283 | 235.7 | 78.7 | 20.7 | 72.4 ± 1.0 |
| FITPRUNE(AAAI25) | ✗ | 192 | 219 | 199.5 | 77.4(×1.02) | 20.4 | 34.1 ± 0.8 |
| NÜWA(ICLR26) | ✗ | 192 | 219 | 181.5 | 75.4(×1.04) | 20.4 | 23.4 ± 0.6 |
| *RTPrune* | ✗ | 192 | 219 | 181.5 | 72.6(×1.08) | 20.4 | 57.7 ± 1.0 |
| DIVPRUNE(CVPR25) | ✔ | AVG=213 | AVG=240 | 199.2 | 75.1(×1.05) | 20.5 | 50.1 ± 1.1 |
| CDPRUNER(NIPS25) | ✔ | AVG=213 | AVG=240 | 199.2 | 75.1(×1.05) | 20.5 | 53.0 ± 1.1 |
| *RTPrune* | ✔ | AVG=213 | AVG=240 | 199.2 | 75.1(×1.05) | 20.5 | 63.7 ± 1.1 |
| | | | DEEPSEEK-OCR-LARGE | | | | |
| BASELINE | - | 400 | 431 | 363.0 | 96.3 | 20.9 | 75.5 ± 1.0 |
| FASTV(ECCV24) | ✗ | 300 | 331 | 290.9 | 92.9(×1.04) | 20.6 | 25.2 ± 0.5 |
| DART(EMNLP25) | ✗ | 300 | 331 | 290.9 | 92.9(×1.04) | 20.6 | 25.6 ± 0.5 |
| *RTPrune* | ✗ | 300 | 331 | 276.6 | 77.1(×1.25) | 20.6 | 68.3 ± 1.1 |
| SPARSEVLM(ICML25) | ✔ | AVG=336 | AVG=363 | 333.5 | 92.4(×1.04) | 20.7 | 32.0 ± 0.8 |
| VISIONZIP(CVPR25) | ✔ | AVG=336 | AVG=367 | 307.5 | 78.1(×1.23) | 20.7 | 59.7 ± 1.1 |
| *RTPrune* | ✔ | AVG=336 | AVG=367 | 307.5 | 78.1(×1.23) | 20.7 | 73.9 ± 1.1 |

*Table 5.* Performance comparison of various pruning methods on OmniDocBench using various OCR models.

| METHOD | #TOKENS | PREFILL | DECODE | TEXT$^{EDIT}$ ↓ | FORMULA$^{CDM}$ ↑ | TABLE$^{TEDS/TEDS-S}$ ↑ | ORDER$^{EDIT}$ ↓ | OVERALL ↑ |
|---|---|---|---|---|---|---|---|---|
| | | | DEEPSEEK-OCR2-GUNDAM | | | | | |
| BASELINE | 1083 | 59.1 | 16.8 | 0.05 | 90.54 | 86.94 / 91.23 | 0.06 | 90.93 |
| FITPRUNE(AAAI25) | AVG=560 | 58.3 | 15.9 | 0.22 | 65.89 | 62.23 / 66.67 | 0.20 | 68.64 |
| DIVPRUNE(CVPR25) | AVG=560 | 57.6 | 16.6 | 0.27 | 65.77 | 57.67 / 68.91 | 0.19 | 65.35 |
| CDPRUNER(NIPS25) | AVG=560 | 57.6 | 16.6 | 0.45 | 50.30 | 39.61 / 52.79 | 0.30 | 48.37 |
| *RTPrune* | AVG=560 | 57.6 | 16.6 | 0.14 | 85.07 | 80.44 / 85.95 | 0.10 | **83.94** |
| | | | LIGHTONOCR | | | | | |
| BASELINE | 2137 | 30.4 | 16.4 | 0.16 | 87.00 | 83.22 / 89.49 | 0.10 | 84.91 |
| FITPRUNE(AAAI25) | AVG=1238 | 27.8 | 16.1 | 0.94 | 6.33 | 4.92 / 7.14 | 0.65 | 5.85 |
| DIVPRUNE(CVPR25) | AVG=1238 | 23.4 | 16.0 | 0.29 | 70.17 | 62.89 / 76.76 | 0.19 | 67.95 |
| CDPRUNER(NIPS25) | AVG=1238 | 23.4 | 16.0 | 0.22 | 88.45 | 73.42 / 84.58 | 0.13 | 79.96 |
| *RTPrune* | AVG=1238 | 23.4 | 16.0 | 0.18 | 85.67 | 81.26 / 87.53 | 0.12 | **83.00** |
| | | | GLM-OCR | | | | | |
| BASELINE | 4512 | 26.3 | 11.0 | 0.10 | 86.05 | 13.92 / 15.29$^{‡}$ | 0.11 | 63.22 |
| FITPRUNE(AAAI25) | AVG=2783 | 18.3 | 10.7 | 0.49 | 57.62 | 13.65 / 15.81 | 0.37 | 40.92 |
| DIVPRUNE(CVPR25) | AVG=2783 | 17.1 | 10.7 | 0.57 | 41.92 | 1.82 / 2.47 | 0.46 | 28.95 |
| CDPRUNER(NIPS25) | AVG=2783 | 17.1 | 10.7 | 0.16 | 77.31 | 5.15 / 5.58 | 0.17 | 55.39 |
| *RTPrune* | AVG=2783 | 17.1 | 10.7 | 0.15 | 81.74 | 12.72 / 13.91 | 0.14 | **59.98** |

Gundam mode of DeepSeek-OCR, we conduct experiments on the Ocean-OCR benchmark, which includes document extraction, scene text recognition, and handwritten recognition. The results in Figure 5 demonstrate that *RTPrune* not only outperforms existing pruning methods by a substantial margin but also achieves performance comparable to, and in some cases surpassing, the unpruned baseline. Crucially, together with the consistent improvements observed across other DeepSeek-OCR modes, the strong performance on the Gundam mode validates the robust generalization capability of our approach, showing that *RTPrune* consistently delivers optimal accuracy across all modes of the DeepSeek-OCR.

## 4.3. Efficiency Analysis

To demonstrate the efficiency of *RTPrune*, we conduct a comparative analysis against other pruning methods in terms of the number of tokens, GFLOPs, total prefill time (ms) and decode time (ms/token) on the olmOCR-Bench. As shown

in Table 4, compared to all other pruning methods, *RTPrune* consistently achieves the best efficiency while maintaining the highest performance. We find that the limited latency reduction in within-LLM and multi-stage pruning methods primarily stems from frequent tensor shape variations, thereby degrading hardware throughput and offsetting the theoretical gains from token reduction. In contrast, *RTPrune* performs token pruning before the prefill stage and explicitly leverages the connection between the vision encoder and the LLM decoding process, thereby outperforming other methods that prune at the vision encoder. Furthermore, *RT-Prune* with a dynamic pruning ratio strategy maintains a performance profile nearly identical to the dense baseline, demonstrating an optimal trade-off between computational cost and recognition accuracy.

---

$^{‡}$The `transformers`-based GLM-OCR outputs plain text tables instead of OmniDocBench-required HTML/LaTeX, deflating its scores.

*Table 6.* Ablation study on our selection metric and optimal token merging (OTM). Text and Read Order are measured by *Edit Distance↓*; Formula is measured by *CDM (Character Distribution Match)↑*; Table is measured by *TEDS (Tree Edit Distance-based Similarity)* ↑ and *TEDS-S↑*.

| METHODS | TEXT | FORMULA | TABLE | ORDER | OVERALL |
|---|---|---|---|---|---|
| DEEPSEEK-OCR-BASE | 0.12 | 81.90 | 84.58 / 88.43 | 0.11 | 84.83 |
| VARIANCE | 0.20 | 76.40 | 78.33 / 83.46 | 0.19 | 78.21 |
| +OTM | 0.19 | 76.91 | 79.51 / 84.49 | 0.18 | 79.20 |
| ENTROPY | 0.53 | 61.83 | 51.99 / 63.89 | 0.29 | 53.61 |
| +OTM | 0.53 | 63.40 | 54.09/66.43 | 0.29 | 54.96 |
| $\ell_2$-NORM | 0.18 | 79.00 | 80.18 / 85.32 | 0.17 | 80.53 |
| +GTP-VIT(WACV24) | 0.18 | 79.33 | 80.35 / 85.11 | 0.17 | 80.59 |
| +VISIONZIP(CVPR25) | **0.17** | **80.14** | 80.15 / 84.77 | **0.16** | 81.03 |
| +SPARSEVLM(ICML25) | 0.18 | 79.51 | 80.26 / 85.02 | 0.17 | 80.69 |
| +NÜWA(ICLR26) | 0.17 | 79.32 | 80.79 / 85.67 | 0.17 | 80.90 |
| +OTM(*RTPrune*) | **0.17** | 79.95 | **81.60 / 86.74** | **0.16** | **81.48** |

*Table 7.* Ablation study on dynamic pruning ratio. $r$ refers to the pruning ratio.

| METHODS | DOCUMENT | | SCENE TEXT | HANDWRITING | |
|---|---|---|---|---|---|
| | EN | ZH | | EN | ZH |
| DEEPSEEK-OCR-BASE | 90.89 | 92.63 | 53.26 | 67.55 | 81.72 |
| AVG $r_{\text{DYN}}$ | **0.18** | **0.18** | **0.17** | **0.30** | **0.27** |
| FITPRUNE(AAAI25) | 63.32 | 90.80 | 46.86 | 33.51 | 71.51 |
| CDPRUNER(NIPS25) | 62.49 | 63.63 | 55.44 | 64.16 | 83.43 |
| NÜWA(ICLR26) | 2.12 | 1.34 | 35.81 | 19.91 | 34.33 |
| *RTPrune* | 82.29 | 84.46 | 56.11 | 66.07 | 83.59 |
| FIXED $r$ | | | **0.22** | | |
| FITPRUNE(AAAI25) | 62.06 | 90.58 | 45.51 | 32.93 | 70.89 |
| CDPRUNER(NIPS25) | 55.06 | 57.83 | 54.40 | 66.24 | 82.66 |
| NÜWA(ICLR26) | 1.67 | 1.32 | 35.25 | 19.84 | 33.24 |
| *RTPrune* | 74.39 | 79.62 | 56.06 | 66.13 | 82.92 |

### 4.4. Generalizability

With the growing emergence of end-to-end OCR models (e.g. DeepSeek-OCR2 (Wei et al., 2026), LightOnOCR (Taghadouini et al., 2026), GLM-OCR(Duan et al., 2026)), we expect the phenomenon of text-rich tokens having larger norms to generalize beyond DeepSeek-OCR. To verify that our method is not narrowly tailored to DeepSeek-OCR, we further evaluate its generalizability on several recently released end-to-end OCR models with OmniDocBench. Additional experiments in Table 5 demonstrate that while other pruning methods exhibit significant performance fluctuations across different OCR models, our approach remains remarkably stable, preserving over 92% of the original performance across all tested architectures. Furthermore, our method delivers substantial inference acceleration with minimal degradation across key metrics, firmly validating both its effectiveness and broad generalizability.

### 4.5. Ablation Study

**Ablation on selection metric.** Table 6 evaluates the performance of $\ell_2$-norm selection metric with or without our optimal token merging, including variance and entropy, on OmniDocBench using DeepSeek-OCR-Base. $\ell_2$-norm selection metric performs better than other importance metrics. Variance-based selection also yields strong results, since features with larger norms often exhibit a wider value distribution and thus higher variance, but may miss tokens whose norm increases mainly due to an overall feature shift rather than larger dispersion.

**Ablation on token merging.** Table 6 evaluates the performance of norm-based pruning integrated with various token merging techniques, including those from GTP-ViT (Xu et al., 2024) and related token pruning approaches, on OmniDocBench using DeepSeek-OCR-Base. All token merging methods consistently improve OCR performance, indirectly confirming the reading twice behavior of the model.

Among them, our method achieves the best overall results, demonstrating the effectiveness of optimal transport–based feature matching for visual token propagation.

**Ablation on dynamic pruning ratio.** Table 7 reports the precision of various pruning methods with and without the dynamic pruning strategy on the Ocean-OCR benchmark using DeepSeek-OCR-Base. Our dynamic pruning strategy accurately captures task-dependent information density, recognizing that handwritten recognition typically contains less informative content than document extraction and scene text recognition, and thus enabling differentiated pruning behaviors across tasks. Under the same average pruning ratio, the proposed strategy yields up to 13.5% higher accuracy than fixed-ratio pruning in scenarios with higher adaptive pruning rates (e.g., document extraction and scene text recognition), while maintaining comparable accuracy in cases with lower adaptive pruning rates (e.g., handwritten recognition). These results demonstrate that our content-adaptive pruning facilitates a more optimal balance between efficiency and accuracy.

## 5. Conclusion

In this paper, we propose *RTPrune*, a training-free visual token pruning method tailored for accelerating DeepSeek-OCR inference. By analyzing the decoding process, we identify a two-stage attention pattern over visual tokens and accordingly design a two-stage pruning strategy that preserves high-norm tokens and merges the remaining ones via optimal transport. We further introduce a dynamic pruning ratio that adapts to token similarity and textual density, enabling an improved efficiency–accuracy trade-off for OCR tasks. Extensive experiments demonstrate that *RTPrune* achieves state-of-the-art performance across multiple OCR benchmarks. Efficiency analysis further shows *RTPrune* substantially reduces inference latency while preserving accuracy, providing a robust foundation for future research and deployment of visual-text compression.

## Impact Statement

DeepSeek-OCR's visual text compression encodes text as visual tokens, achieving up to a 10× reduction in sequence length and substantially lowering the computational cost of long-document processing for large language models. In this paper, we propose a token pruning method tailored to DeepSeek-OCR that further shortens the inference sequence while maintaining stable OCR accuracy, significantly improving inference speed and enabling secondary compression of visual text representations. Despite these efficiency gains, we urge practitioners to exercise caution when deploying token compression in high-stakes domains (e.g., legal or medical scenes), where even minor information loss during pruning could yield severe real-world consequences.

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

Section A provides a comprehensive analysis of the inherent limitations in existing token pruning methodologies, serving as a motivation for our approach. Section B describes the sobel operator used in dynamic pruning ratio and Section C describes sinkhorn algorithm used in token merging. Section D provides the detailed algorithm of our method *RTPrune*. Section E provides some details of the experimental setup, including information about model architecture of DeepSeek-OCR, evaluation benchmarks and comparison methods. Section F presents additional experimental results. Section G discusses the limitations and future directions of this work.

## A. Detailed Discussions for Limitations of Existing Methods in DeepSeek-OCR

### A.1. Redundancy of Visual Tokens

To investigate visual tokens redundancy within DeepSeek-OCR, we performed experiments of pruning a single token across selected images. Figure 6 demonstrates that removing a single visual token has no impact on the generated text. Statistically, for over 95% of the visual tokens, their individual removal does not prevent the model from generating identical and correct outputs. This stems from the fact that the textual and structural information stored in most tokens is not unique, thereby providing strong evidence of visual tokens redundancy in DeepSeek-OCR.

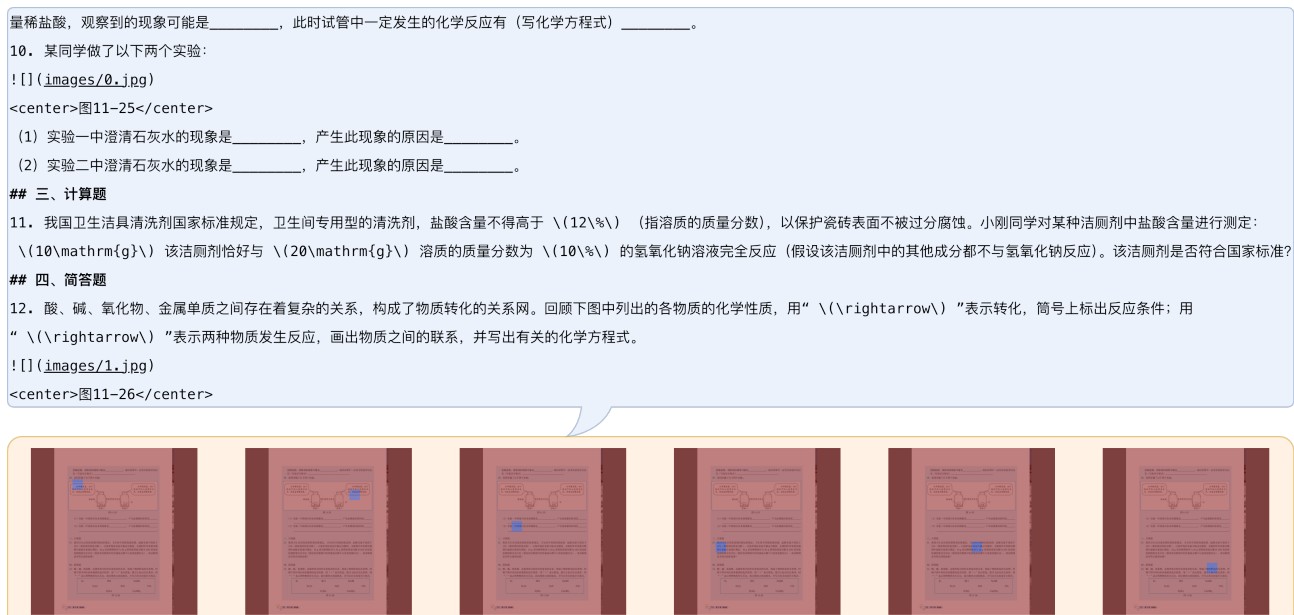

*Figure 6.* Visualizations of visual tokens redundancy on DeepSeek-OCR-Base. The patch highlighted in blue is pruned and the patches highlighted in red are kept.

### A.2. LLM Attention-based Methods

To explore the significance of visual tokens in terms of the attention they receive within the language model, we conducted experiments by pruning the bottom 25% of visual tokens based on attention scores at each layer. As shown in Table 8, pruning within the first nine layers leads to a substantial degradation in OCR accuracy and favorable results are only observed when pruning occurs in the final three layers.

Unlike VQA tasks, OCR requires the LLM to attend to a broader range of visual tokens to capture comprehensive textual and structural information, leading to diverse attention patterns across different layers. While pruning at deeper layers preserves model accuracy by leveraging complete information from preceding layers, the resulting inference speedup is negligible. Even with multi-stage pruning within the LLM (Zhang et al., 2024a), most existing methods concentrate pruning efforts in the shallower layers due to inference speed constraints. However, this strategy fails to capture the critical visual tokens prioritized by the deeper layers. This observation motivates us to further investigate the relationship between visual encoder embeddings and the attention mechanisms within the LLM.

*Table 8.* Performance comparison of 25% pruning across each layer of the LLM in DeepSeek-OCR on OmniDocBench (10% subset).

| PRUNED LAYER | #TOKENS | GFLOPs | TEXT$^{EDIT}$ ↓ | FORMULA$^{CDM}$ ↑ | TABLE$^{TEDS}$ ↑ | TABLE$^{TEDS-S}$ ↑ | ORDER$^{EDIT}$ ↓ | OVERALL ↑ |
|---|---|---|---|---|---|---|---|---|
| | | | | DEEPSEEK-OCR-BASE | | | | |
| BASELINE | 256 | 235.7 | 0.177 | 83.47 | 88.822 | 92.735 | 0.160 | 84.864 |
| 0 | | 185.9 | 0.959 | 0.000 | -0.120 | 1.334 | 0.602 | 1.327 |
| 1 | | 190.4 | 0.942 | 1.065 | 0.332 | 0.968 | 0.654 | 2.399 |
| 2 | | 194.9 | 0.904 | 5.019 | -0.286 | 2.273 | 0.626 | 4.778 |
| 3 | | 199.5 | 0.878 | 17.199 | 6.438 | 10.939 | 0.635 | 11.946 |
| 4 | | 204.0 | 0.853 | 10.967 | 8.663 | 16.073 | 0.597 | 11.443 |
| 5 | 192 | 208.5 | 0.847 | 3.139 | 10.073 | 12.544 | 0.618 | 9.504 |
| 6 | | 213.1 | 0.775 | 8.273 | 11.895 | 14.170 | 0.574 | 14.223 |
| 7 | | 217.6 | 0.585 | 23.075 | 18.238 | 20.307 | 0.486 | 27.604 |
| 8 | | 222.1 | 0.494 | 46.937 | 45.595 | 48.523 | 0.418 | 47.711 |
| 9 | | 226.6 | 0.315 | 67.803 | 69.744 | 73.915 | 0.296 | 68.682 |
| 10 | | 231.2 | 0.235 | 76.996 | 77.771 | 82.260 | 0.206 | 77.089 |
| 11 | | 235.7 | 0.209 | 72.178 | 80.616 | 84.339 | 0.188 | 77.298 |
| | | | | DEEPSEEK-OCR-LARGE | | | | |
| BASELINE | 400 | 363.0 | 0.138 | 77.241 | 84.194 | 87.556 | 0.101 | 82.545 |
| 0 | | 283.6 | 0.959 | 11.453 | 0.115 | 2.593 | 0.593 | 5.223 |
| 1 | | 290.7 | 0.963 | 0.000 | 0.130 | 2.243 | 0.654 | 1.277 |
| 2 | | 297.8 | 0.875 | 7.875 | -0.516 | 1.718 | 0.648 | 6.620 |
| 3 | | 305.0 | 0.852 | 14.304 | 5.387 | 11.351 | 0.586 | 11.497 |
| 4 | | 312.1 | 0.813 | 6.444 | 13.923 | 18.352 | 0.587 | 13.022 |
| 5 | 300 | 319.3 | 0.798 | 2.142 | 12.845 | 15.343 | 0.567 | 11.729 |
| 6 | | 326.4 | 0.698 | 5.284 | 19.008 | 22.205 | 0.509 | 18.164 |
| 7 | | 333.6 | 0.598 | 43.761 | 32.753 | 37.760 | 0.473 | 38.905 |
| 8 | | 340.7 | 0.462 | 48.373 | 45.072 | 46.129 | 0.380 | 49.082 |
| 9 | | 347.8 | 0.306 | 58.550 | 60.926 | 65.681 | 0.293 | 62.959 |
| 10 | | 355.0 | 0.188 | 64.982 | 82.668 | 85.176 | 0.161 | 76.283 |
| 11 | | 362.1 | 0.159 | 67.417 | 85.693 | 88.186 | 0.170 | 79.070 |

## A.3. Textual Relevance-based Methods

The vision encoder in DeepSeek-OCR, referred to as DeepEncoder, is comprised of SAM (Kirillov et al., 2023) and CLIP-ViT-B/16 (Radford et al., 2021) models. Following visualization in CDPruner (Zhang et al., 2025b), Figure 7 visualizes the correlation between the input prompt `text` and the image embeddings processed by different CLIP. Compared to CLIP-ViT-L/14-336px, CLIP-ViT-B/16 inherently exhibits weaker multi-modal alignment. This alignment is further attenuated as the CLIP model within DeepEncoder has undergone re-training. Moreover, CLIP primarily provides spatial localization for the `text` prompt rather than capturing precise fine-grained textual information.

Furthermore, while prompts in VQA tasks are dynamic—meaning the output depends heavily on the relevance to a specific query—prompts in OCR tasks are typically invariant, such as: `<image>\n<|grounding|>Convert the document to markdown.` Consequently, the relevance scores between the fixed prompt and visual tokens lack the necessary discriminative power to identify essential information, as the prompt itself does not change across different images. This limitation prevents us from using relevance-based metrics for token filtering and motivates us to seek alternative methodologies.

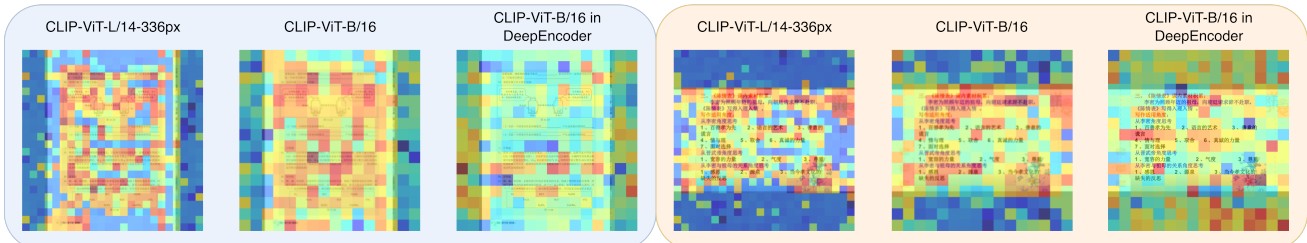

*Figure 7.* Visualizations of textual relevance in different CLIP model. The color gradient indicates the degree of relevance, where cooler blue tones represent lower correlation and warmer red tones denote higher relevance.

## A.4. Similarity-based Methods

Recently, diversity-based approaches (Alvar et al., 2025; Wen et al., 2025; Zhang et al., 2025b) have gained significant traction in VLM pruning research. However, in OCR tasks, an indiscriminate pursuit of visual token diversity can be counterproductive. Such methods may prioritize tokens representing background or layout structures at the expense of essential textual content, leading to a decline in OCR performance. Consequently, similarity-based diversity metrics are sub-optimal for DeepSeek-OCR, necessitating the exploration of alternative pruning strategies that better preserve textual fidelity.

## B. Sobel Operator for Edge Detection

The Sobel operator is a discrete differential operator used to compute an approximation of the gradient of the image intensity function. In the context of document analysis, it is particularly effective at identifying character strokes and boundaries, which exhibit sharp intensity transitions against the background.

Firstly, the input RGB image is converted to a grayscale image $\mathbf{I}$ by calculating a weighted sum of the red ($R$), green ($G$), and blue ($B$) channels. This process is formally defined as:

$$\mathbf{I}(i, j) = w_R \cdot R(i, j) + w_G \cdot G(i, j) + w_B \cdot B(i, j), \tag{10}$$

where $(i, j)$ denotes the pixel coordinates. Following the standard ITU-R BT.601 (BT et al., 2011) coefficients for terrestrial broadcasting, the weights are typically set as $w_R = 0.299$, $w_G = 0.587$, and $w_B = 0.114$. This ensures that the resulting grayscale representation preserves the perceptual contrast necessary for robust edge detection via the Sobel operator.

For an input grayscale image $\mathbf{I}$, the operator utilizes two $3 \times 3$ kernels convolved with the original image to calculate approximations of the horizontal and vertical derivatives:

$$\mathbf{G}_x = \begin{bmatrix} -1 & 0 & +1 \\ -2 & 0 & +2 \\ -1 & 0 & +1 \end{bmatrix} * \mathbf{I}, \quad \mathbf{G}_y = \begin{bmatrix} +1 & +2 & +1 \\ 0 & 0 & 0 \\ -1 & -2 & -1 \end{bmatrix} * \mathbf{I}, \tag{11}$$

where $*$ denotes the 2-D convolution operation. At each pixel coordinate $(i, j)$, the gradient magnitude $M(i, j)$ is computed as:

$$G(i, j) = \sqrt{G_x(i, j)^2 + G_y(i, j)^2}. \tag{12}$$

In our *RTPrune* framework, the Sobel operator serves as the foundation for quantifying the **Textual Density ($\rho$)**. Unlike natural scenes, document images are composed of sparse but high-contrast signals (text, image, table) and uniform regions (background).

By applying a global threshold $\tau$ to the gradient magnitude $G$, we generate a binary edge map $E$:

$$E(i, j) = \begin{cases} 1, & \text{if } G(i, j) \geq \tau \\ 0, & \text{otherwise} \end{cases}. \tag{13}$$

The density of a specific visual token (or patch) is determined by the ratio of "active" pixels within its spatial boundaries. High-density patches typically correspond to complex characters or dense formatting, while low-density patches represent margins or blank spaces. This allows *RTPrune* to adaptively allocate more computational budget to information-rich regions while aggressively pruning redundant background tokens.

## C. Sinkhorn Algorithm for Optimal Token Matching

To facilitate the seamless integration of redundant information into the preserved tokens, we employ the Sinkhorn algorithm to solve the entropy-regularized optimal transport problem. This approach ensures a globally optimized matching between the set of kept tokens $\mathcal{T}_{kept}$ and the set of proposed-to-prune tokens $\mathcal{T}_{prop}$.

Given the kept tokens $\mathbf{T}_k \in \mathbb{R}^{M \times D}$ and proposed tokens $\mathbf{T}_p \in \mathbb{R}^{(N-M) \times D}$, we first compute the cosine similarity matrix $\mathbf{S} \in \mathbb{R}^{M \times (N-M)}$ via batch matrix multiplication of $\ell_2$-normalized features:

$$\mathbf{S} = (\bar{\mathbf{T}}_k)(\bar{\mathbf{T}}_p)^{\top}. \tag{14}$$

To handle tokens that do not find a suitable match, we augment the score matrix with a dustbin row and column using a fixed parameter $z$, following the SuperGlue (Sarlin et al., 2020) paradigm. This augmented matrix $\bar{\mathbf{S}} \in \mathbb{R}^{(M+1) \times (N-M+1)}$ allows the model to suppress irrelevant features during the transport process.

The goal is to find a transport plan $\bar{\mathbf{P}}$ that maximizes the total similarity while satisfying marginal constraints. For numerical stability, the iterations are performed in log-space. Let $\mathbf{Z}$ be the log-augmented score matrix. We define the marginal distributions $\mu$ and $\nu$ as:

$$\log \mu = [\mathbf{0}_M, \log(N - M)], \quad \log \nu = [\mathbf{0}_{N-M}, \log M]. \tag{15}$$

The Sinkhorn algorithm iteratively updates the dual variables $\mathbf{u} \in \mathbb{R}^{M+1}$ and $\mathbf{v} \in \mathbb{R}^{N-M+1}$:

$$\mathbf{u}^{(t+1)} = \log \mu - \text{LogSumExp}(\mathbf{Z} + \mathbf{v}^{(t)\top}), \tag{16}$$

$$\mathbf{v}^{(t+1)} = \log \nu - \text{LogSumExp}(\mathbf{Z}^{\top} + \mathbf{u}^{(t+1)\top}), \tag{17}$$

where $t$ denotes the iteration index. After $T$ iterations ($T = 100$ in our implementation), the final transport matrix in log-space is obtained by $\mathbf{Z}_{\text{final}} = \mathbf{Z} + \mathbf{u}\mathbf{1}^{\top} + \mathbf{1}\mathbf{v}^{\top}$.

The optimal transport plan $\bar{\mathbf{P}}$ is extracted from the non-dustbin quadrant of $\exp(\mathbf{Z}_{\text{final}})$ and the solution $\mathbf{P}$ can be obtained by dropping the dustbins $\mathbf{P} = \bar{\mathbf{P}}_{:-1,:-1}$. The redundant information from $\mathbf{T}_p$ is then aggregated into $\mathbf{T}_k$ using a weighted sum:

$$\mathbf{T}_k = \mathbf{T}_k + \alpha(\mathbf{P} \cdot \mathbf{T}_p), \tag{18}$$

where $\alpha$ is a balancing coefficient. This mechanism ensures that even pruned tokens contribute their salient visual information to the remaining sequence, thereby maintaining OCR accuracy under high compression ratios.

## D. The Algorithm Framework for *RTPrune*

---
**Algorithm 1** *RTPrune*

---
**Input:** original image $\mathbf{I}$, visual token embeddings $\mathbf{T}_v$, fixed parameter of dustbin in Sinkhorn algorithm $z = 0.2$, balancing coefficient $\alpha = 0.1$ and threshold for edge detection $\tau = 0.2$

\# *Optional: calculate pruning ratio.*

calculate the average textual density $\rho$                                                  ▷ Equation (8)

calculate the average embedding similarity $\varphi$                                 ▷ Equation (7)

calculate pruning ratio                                                       ▷ Equation (9)

\# *Stage 1: dominant token selection.*

select $M = N(1 - r)$ visual tokens as $\mathcal{T}_{\text{kept}}$                                  ▷ Equation (1)

\# *Stage 2: optimal token merging.*

calculate the similarity matrix between $\mathcal{T}_{\text{kept}}$ and $\mathcal{T}_{\text{prop}}$

use Sinkhorn algorithm to find an optimal token matching plan $\mathbf{P}$

merge $\mathcal{T}_{\text{prop}}$ to $\mathcal{T}_{\text{kept}}$ with $\mathbf{P}$                                                 ▷ Equation (6)

**Output:** merged visual tokens $\mathcal{T}_{\text{kept}}$

---

## E. Details of Experimental Setup

### E.1. Model Architecture of DeepSeek-OCR

DeepSeek-OCR[*] adopts a unified end-to-end vision-language model (VLM) architecture, consisting of two core components: DeepEncoder (the vision encoder) and DeepSeek3B-MoE-A570M (the language decoder). They are connected by a projector. DeepEncoder serves as the key engine, responsible for extracting high-resolution image features, tokenizing

---
[*]https://github.com/deepseek-ai/DeepSeek-OCR

visual information, and compressing vision tokens while maintaining low activation memory. The decoder, based on a mixture-of-experts (MoE) architecture, reconstructs text representations from compressed vision tokens, activating only 570M parameters during inference to balance expressive power and inference efficiency. This two-component design forms a closed loop of "visual compression-text reconstruction," enabling efficient long-context optical compression.

DeepEncoder, with a total of approximately 380M parameters, is composed of three serially connected modules: an 80M SAM-base for window attention-dominated visual perception, a 16× convolutional compressor (2-layer convolution with kernel size 3, stride 2, and padding 1), and a 300M CLIP-large for dense global attention-based knowledge extraction (with its original patch embedding layer removed). It supports multiple resolution modes to adapt to different compression ratio needs. The SAM-base segments images into patch tokens, the convolutional compressor downsamples tokens by 16× to reduce computational load, and the CLIP-large models long-range dependencies of compressed tokens, collectively achieving efficient vision-text compression.

The language decoder (DeepSeek3B-MoE-A570M) is instantiated based on the DeepSeekV2 architecture, a MoE-optimized LLM tailored for OCR long-context decoding. Structurally, it comprises 12 decoder layers: 1 standard DecoderLayer for initial feature alignment and 11 MoE DecoderLayers for expert-driven computation, with a hidden dimension of 1280 to match DeepEncoder's output. Each MoE layer adopts a top-k=6 expert activation strategy, integrating task-specific experts (intermediate dimension 896) and a shared expert (intermediate dimension 1792), activating only 570M parameters per inference to balance capacity and efficiency. A core optimization lies in dynamic token adaptation: aligned with DeepEncoder's compression ratio, the decoder adjusts token counts layer-wise during decoding, reducing self-attention's $n^2$ complexity. Combined with graph-based token aggregation (utilizing cosine similarity for feature compensation), DeepSeekV2 minimizes GFLOPs and latency while preserving text reconstruction accuracy, perfectly complementing DeepSeek-OCR's "visual compression-text reconstruction" loop for efficient long-context OCR.

To meet both research and practical application requirements, DeepEncoder provides two major types of resolution modes: native resolution and dynamic resolution. The native resolution mode includes four sub-modes: Tiny (512×512, 64 vision tokens), Small (640×640, 100 vision tokens), Base (1024×1024, 256 vision tokens), and Large (1280×1280, 400 vision tokens). For Tiny and Small modes with smaller resolutions, images are directly resized to match the mode's resolution to avoid wasting vision tokens; for Base and Large modes, images are padded to preserve the original aspect ratio. All modes are trained simultaneously to enable a single DeepSeek-OCR model to flexibly adapt to different vision-text compression ratios.

### E.2. Evaluation Benchmarks

**OmniDocBench v1.5**[†] (**Ouyang et al., 2025**): OmniDocBench is a comprehensive benchmark for diverse PDF document parsing, featuring extensive annotations covering multiple document types, languages, and content modalities. It includes both English and Chinese documents, with rich content such as plain text, mathematical formulas, tables, and structured layouts, making it suitable for evaluating practical OCR performance in real-world scenarios. The benchmark adopts edit distance as the core evaluation metric (smaller values indicate better performance), which quantifies the similarity between the model's output and the ground truth by measuring the minimum number of character insertions, deletions, or substitutions required for alignment. It also categorizes documents into sub-types like books, slides, financial reports, academic papers, and newspapers, enabling fine-grained analysis of model performance across different application scenarios. The evaluation assesses the model's accuracy in parsing PDF page content. The evaluation uses the model's Markdown output of the entire PDF page parsing results as the prediction. The Overall metric is calculated as:

$$\text{Overall} = \frac{(1 - \text{Text Edit Distance}) \times 100 + \text{Table TEDS} + \text{Formula CDM}}{3}. \tag{19}$$

**olmOCR-Bench**[‡] (**Poznanski et al., 2025a**): olmOCR-Bench is designed to unlock trillion-scale tokens in PDF documents, focusing on large-scale and diverse document OCR tasks. The benchmark comprises a vast corpus of PDF files with varied layouts, languages, and complexity levels, ranging from simple text documents to complex multi-modal content. It aims to evaluate the scalability and robustness of OCR models in processing massive volumes of unstructured or semi-structured PDF data, which is critical for large-scale LLM/VLM pretraining data generation. Evaluation metrics typically include text recognition accuracy, layout parsing precision, and token-level alignment accuracy, emphasizing both the correctness of text

---

[†]https://github.com/opendatalab/OmniDocBench

[‡]https://github.com/allenai/olmocr/tree/main/olmocr/bench

extraction and the preservation of document structure.

**Ocean-OCR**[§](Chen et al., 2025) **:** Ocean-OCR is a state-of-the-art OCR benchmark tailored for evaluating advanced document understanding capabilities, including complex layout parsing, multi-modal content recognition (e.g., charts, diagrams, and handwritten text), and cross-language OCR performance. The benchmark features a diverse dataset of documents from various domains (e.g., academia, finance, and daily life) with varying resolutions, noise levels, and structural complexities. It adopts a multi-dimensional evaluation framework, including text recognition accuracy, layout detection F1-score, and structured information extraction precision, to comprehensively assess a model's ability to handle real-world OCR challenges. Its design emphasizes the integration of visual perception and language understanding, aligning with the core goals of modern VLM-based OCR systems.

### E.3. Comparison Methods

#### E.3.1. ATTENTION-BASED METHODS

**FastV (**Chen et al., 2024**):** The first work to identify the inefficient visual attention phenomena in MLLMs. Based on this observation, FastV proposes a straightforward solution, that is, to prune the part of visual tokens with the lowest visual-text attention score after layer 2 of the model, thereby achieving MLLM inference acceleration in a training-free manner.

**FitPrune (**Ye et al., 2025**):** FitPrune is a training-free visual token pruning method for MLLMs that frames pruning as attention distribution fitting, jointly considering self- and cross-attention to minimize divergence before and after pruning. It leverages attention statistics from a small batch of data, uses binary search to generate a layer-wise pruning recipe meeting a predefined budget, and ranks tokens by a combined attention-based importance score to prune the least critical ones during inference—no extra modules or gradient computations required.

#### E.3.2. SIMILARITY-BASED METHODS

**DivPrune (**Alvar et al., 2025**):** This work also focuses on token diversity. However, unlike previous approaches, DivPrune reformulates the token pruning problem as a MMDP, aiming to retain the most diverse subset by maximizing the minimum pairwise distance among the selected tokens.

**DART (**Wen et al., 2025**):** This work argues that in token pruning, duplication matters more than importance. Based on this insight, it first selects a small set of pivot tokens, and then iteratively retains the most diverse tokens from the remaining ones by selecting those with the lowest similarity to the already selected tokens. Finally, a group of the most diverse visual tokens is obtained.

#### E.3.3. ATTENTION&SIMILARITY-BASED METHODS

**VisionZip (**Yang et al., 2025**):** VisionZip relies on visual information for token pruning. It observes that attention within the visual encoder is highly concentrated, and therefore first selects several dominant tokens based on visual attention. Then, among all the remaining tokens, a set of contextual tokens is obtained through clustering. These two groups are combined and fed into the language model, aiming to preserve as much visual information as possible.

**NÜWA (**Huang et al., 2026**):** Nüwa is a two-stage token pruning method that enables efficient feature aggregation while maintaining spatial integrity. In the first stage, after the vision encoder, it applies three operations, namely separation, alignment, and aggregation, which are inspired by swarm intelligence algorithms to retain information-rich global spatial anchors. In the second stage, within the LLM, it performs text-guided pruning to retain task-relevant visual tokens.

#### E.3.4. TEXTUAL RELEVANCE&ATTENTION-BASED METHODS

**SparseVLM (**Zhang et al., 2024a**):** SparseVLM adopts a multi-stage token pruning strategy and focuses on the impact of the instruction tokens on vision-language attention. It argues that not all text tokens contribute to the visual token pruning, only those highly relevant to the visual content are important. Therefore, it first selects the text tokens most related to the visual input as raters, and uses their attention to the visual tokens to guide the pruning process, leading to further performance improvements.

---

[§]https://github.com/guoxy25/Ocean-OCR

### E.3.5. TEXTUAL RELEVANCE&SIMILARITY-BASED METHODS

**CDPruner ([Zhang et al., 2025b](#)):** CDPruner is a plug-and-play and model-agnostic solution for visual token pruning that maximizes conditional diversity. It first defines the conditional similarity between visual tokens conditioned on the instruction, and then reformulates the token pruning problem with determinantal point process (DPP) to maximize the conditional diversity of the selected subset.

## F. Additional experimental results

### F.1. Ablation Study on Fixed Parameter in Sinkhorn Algorithm

*Table 9.* Performance comparison of 25% pruning across different $z$ in Equation (4) on OmniDocBench.

| METHOD | $z$ | #TOKENS | TEXT$^{\text{EDIT}}$ ↓ | FORMULA$^{\text{CDM}}$ ↑ | TABLE$^{\text{TEDS}}$ ↑ | TABLE$^{\text{TEDS-S}}$ ↑ | ORDER$^{\text{EDIT}}$ ↓ | OVERALL ↑ |
|---|---|---|---|---|---|---|---|---|
| | | | | DEEPSEEK-OCR-BASE | | | | |
| BASELINE | - | 256 | 0.18 | 83.47 | 88.82 | 92.74 | 0.16 | 84.86 |
| *RTPrune* | 0.0 | 192 | 0.20 | 75.92 | 75.67 | 81.98 | 0.19 | 77.06 |
| | 0.2 | | 0.21 | 77.27 | 75.55 | 81.88 | 0.19 | 77.37 |
| | 0.4 | | 0.21 | 76.77 | 75.49 | 81.58 | 0.19 | 77.26 |
| | 0.6 | | 0.20 | 76.09 | 75.23 | 81.45 | 0.19 | 77.04 |
| | 0.8 | | 0.21 | 76.12 | 75.23 | 81.60 | 0.19 | 76.91 |
| | 1.0 | | 0.22 | 76.31 | 74.80 | 81.21 | 0.20 | 76.44 |

In Equation (4), $z$ represents the similarity between visual tokens and the dustbin within our optimal transport framework. A value of $z$ approaching 1 indicates high similarity to the dustbin, suggesting tokens are likely to be discarded without recovery. Conversely, as $z$ approaches 0, tokens are more likely to be matched and merged with other informative tokens. As shown in Table 9, performance degrades significantly as $z$ nears 1, which means a regime that effectively bypasses the merging mechanism in favor of simple pruning. This decline underscores the importance of merging for preserving essential information. Given the model's consistent accuracy across a broad range of $z$ (from 0 to 0.6), which demonstrates the framework's robustness, we adopt $z = 0.2$ as our default.

### F.2. Ablation Study on Merge Strength in Optimal Token Merging

*Table 10.* Performance comparison of 25% pruning across different $\alpha$ in Equation (6) on OmniDocBench.

| METHOD | $\alpha$ | #TOKENS | TEXT$^{\text{EDIT}}$ ↓ | FORMULA$^{\text{CDM}}$ ↑ | TABLE$^{\text{TEDS}}$ ↑ | TABLE$^{\text{TEDS-S}}$ ↑ | ORDER$^{\text{EDIT}}$ ↓ | OVERALL ↑ |
|---|---|---|---|---|---|---|---|---|
| | | | | DEEPSEEK-OCR-BASE | | | | |
| BASELINE | - | 256 | 0.18 | 83.47 | 88.82 | 92.74 | 0.16 | 84.86 |
| *RTPrune* | 0.0 | 192 | 0.22 | 76.31 | 74.80 | 81.21 | 0.20 | 76.44 |
| | 0.05 | | 0.20 | 77.15 | 75.20 | 81.43 | 0.19 | 77.32 |
| | 0.1 | | 0.21 | 77.27 | 75.55 | 81.88 | 0.19 | 77.37 |
| | 0.2 | | 0.20 | 78.01 | 74.87 | 81.21 | 0.19 | 77.36 |
| | 0.5 | | 0.21 | 75.90 | 75.45 | 81.81 | 0.20 | 76.68 |

In Equation (6), $\alpha$ represents the intensity of token merging within our lightweight pruning framework, with higher values signifying more aggressive merging. As shown in Table 10, performance degrades notably at $\alpha = 0$, whereas values between 0.05 and 0.2 yield comparable and superior results. In contrast, when $\alpha$ is increased to 0.5, the formula-level accuracy exhibits the most pronounced decline, which consequently leads to a noticeable drop in overall performance. These observations suggest that moderate merging is beneficial: the $\alpha = 0$ baseline fails to leverage information consolidation, while overly aggressive merging may disrupt the original token-level information and fine-grained textual structures. This performance gap underscores the necessity of merging to preserve essential information that would otherwise be lost through simple pruning. Furthermore, the model's stability across a broad range of $\alpha$ values demonstrates its robustness, leading us to select $\alpha = 0.1$ as the default setting in all experiments.

### F.3. Ablation Study on the Threshold in Sobel Operator

In Equation (8), $\tau$ represents the sensitivity of edge detection, defining the minimum gradient magnitude required for a pixel to be recognized as an informative edge. A higher $\tau$ imposes a more stringent criterion, retaining only salient structural

*Table 11.* Performance comparison of dynamic pruning across different $\tau$ in Equation (8) on OmniDocBench.

| METHOD | $\tau$ | #TOKENS | TEXT$^{\text{EDIT}}$ ↓ | FORMULA$^{\text{CDM}}$ ↑ | TABLE$^{\text{TEDS}}$ ↑ | TABLE$^{\text{TEDS-S}}$ ↑ | ORDER$^{\text{EDIT}}$ ↓ | OVERALL ↑ |
|---|---|---|---|---|---|---|---|---|
| | | | | DEEPSEEK-OCR-BASE | | | | |
| BASELINE | - | 256 | 0.18 | 83.47 | 88.82 | 92.74 | 0.16 | 84.86 |
| | 0.1 | AVG=216 | 0.16 | 80.64 | 81.53 | 86.23 | 0.15 | 81.92 |
| *RTPrune* | 0.2 | AVG=214 | 0.17 | 79.95 | 81.60 | 86.74 | 0.16 | 81.48 |
| | 0.3 | AVG=213 | 0.17 | 79.34 | 81.59 | 86.75 | 0.16 | 81.31 |
| | 0.5 | AVG=213 | 0.17 | 79.79 | 81.39 | 86.19 | 0.16 | 81.36 |

features and resulting in sparser detected text regions, which in turn increases the dynamic pruning ratio and slightly degrades recognition accuracy. Conversely, a lower $\tau$ relaxes the detection threshold, leading to denser edge responses and a reduced pruning ratio, but with a higher risk of preserving redundant background information. From Table 11, the overall performance remains relatively stable across a wide range of $\tau$ values, indicating the robustness of our method to this hyperparameter. Therefore, $\tau$ can be flexibly chosen within a reasonable range, and we empirically set $\tau = 0.2$ in all experiments.

### F.4. Ablation Study on the Pruning Ratio

*Table 12.* Performance comparison of various pruning ratio on OmniDocBench (10% subset).

| METHOD | PRUNING RATIO | #TOKENS | TEXT$^{\text{EDIT}}$ ↓ | FORMULA$^{\text{CDM}}$ ↑ | TABLE$^{\text{TEDS}}$ ↑ | TABLE$^{\text{TEDS-S}}$ ↑ | ORDER$^{\text{EDIT}}$ ↓ | OVERALL ↑ |
|---|---|---|---|---|---|---|---|---|
| | | | | DEEPSEEK-OCR-BASE | | | | |
| BASELINE | 0 | 256 | 0.18 | 83.47 | 88.82 | 92.74 | 0.16 | 84.86 |
| DIVPRUNE | 0.25 | 192 | 0.36 | 70.67 | 61.14 | 72.33 | 0.25 | 65.30 |
| (CVPR25) | 0.5 | 128 | 0.61 | 47.93 | 34.93 | 50.22 | 0.39 | 40.62 |
| *RTPrune* | 0.25 | 192 | 0.23 | 78.42 | 82.04 | 88.89 | 0.22 | 79.15 |
| | 0.5 | 128 | 0.41 | 49.59 | 47.09 | 59.72 | 0.37 | 51.76 |
| | | | | DEEPSEEK-OCR-LARGE | | | | |
| BASELINE | 0 | 400 | 0.14 | 77.24 | 84.19 | 87.56 | 0.10 | 82.55 |
| DIVPRUNE | 0.25 | 300 | 0.27 | 64.34 | 69.05 | 78.10 | 0.19 | 68.83 |
| (CVPR25) | 0.5 | 200 | 0.50 | 51.94 | 46.51 | 67.77 | 0.33 | 49.38 |
| *RTPrune* | 0.25 | 300 | 0.17 | 80.17 | 89.05 | 92.89 | 0.16 | 84.17 |
| | 0.5 | 200 | 0.32 | 64.58 | 68.06 | 76.59 | 0.27 | 66.88 |

While conventional VLMs performing VQA tasks can often sustain high token pruning ratios, DeepSeek-OCR requires more conservative pruning due to the high information density inherent in OCR tasks. As shown in Table 12, evaluation on a 10% subset of OmniDocBench reveals that excessive pruning leads to severe accuracy degradation. Our dynamic pruning strategy addresses this by identifying an optimal equilibrium between efficiency and accuracy. Furthermore, to demonstrate the robustness of our approach across various compression levels, we provide additional results using a fixed 25% pruning rate, which we identify as the near-limit of acceptable accuracy loss.

### F.5. Efficiency Analysis

To evaluate the computational complexity of the MoE-enhanced DeepseekV2 network, it is essential to decompose the floating-point operations (FLOPs) by its core modular components: the self-attention mechanism, the standard feed-forward network (FFN), and the mixture-of-experts (MoE) FFN. We derive the total FLOPs step by step, with consistent notation for all network hyperparameters, and finally aggregate the FLOPs of individual decoder layers to obtain the global formula.

**Step 1: FLOPs of a Single Self-Attention Layer:** The self-attention layer is the core of each decoder layer, consisting of Q/K/V projection, attention score calculation, value weighting, and output projection. For a single self-attention layer, the FLOPs account for all core matrix multiplications (ignoring negligible element-wise operations like rotary embedding and Softmax) and can be expressed as:

$$\text{Attn FLOPs} = \left(8nd^2 + 4n^2 d\right), \tag{20}$$

where $n$ is the sequence length (token number), and $d$ represents the hidden state size. The terms $8nd^2$ and $4n^2 d$ are derived from the four key matrix multiplications of self-attention: Q/K/V projection (3 linear layers), attention score computation $(QK^\top)$, value weighting (score-V multiplication), and output projection (1 linear layer).

**Step 2: FLOPs of a Single Standard FFN:** The standard FFN (used in the non-MoE decoder layer) is composed of **three linear layers** (gate-proj, up-proj, down-proj) with an element-wise activation. Each linear layer contributes $2Bndm$ FLOPs (multiplication plus addition for matrix operation), leading to a total FLOPs for the standard FFN of:

$$\text{Standard FFN FLOPs} = 6ndm, \tag{21}$$

where $m$ is the intermediate size of the standard FFN. The coefficient 6 comes from the sum of FLOPs of the three linear layers ($2ndm + 2ndm + 2nmd$).

**Step 3: FLOPs of a Single MoE FFN:** The MoE FFN consists of $k$ top-activated sparse experts and one universally shared expert (all tokens pass through the shared expert). A single sparse expert has the same three-linear-layer structure as the standard FFN with intermediate size $m_1$, and the shared expert has intermediate size $m_2$. The FLOPs of the MoE FFN are the sum of FLOPs of $k$ activated sparse experts and the shared expert:

$$\text{MoE FFN FLOPs} = 6nd(km_1 + m_2), \tag{22}$$

where $k$ is the number of top-activated experts in MoE, $m_1$ is the intermediate size of a single sparse MoE expert, and $m_2$ is the intermediate size of the shared expert. The coefficient 6 is retained for both sparse and shared experts, consistent with the standard FFN's linear layer composition.

**Step 4: FLOPs of a Single Standard DecoderLayer:** A standard DecoderLayer (no MoE) is a combination of **one self-attention layer** and **one standard FFN**, with FLOPs simply the sum of the two components:

$$\text{Standard Layer FLOPs} = \left(8nd^2 + 4n^2d\right) + 6ndm. \tag{23}$$

In the DeepseekV2 network, the number of such standard decoder layers is denoted as $T_1 = 1$.

**Step 5: FLOPs of a Single MoE DecoderLayer:** A MoE DecoderLayer is a combination of **one self-attention layer** and **one MoE FFN**, with its FLOPs given by the sum of the self-attention FLOPs and the MoE FFN FLOPs:

$$\text{MoE Layer FLOPs} = \left(8nd^2 + 4n^2d\right) + 6nd(km_1 + m_2). \tag{24}$$

In the DeepseekV2 network, the number of such MoE decoder layers is denoted as $T_2 = 11$.

**Step 6: Total FLOPs of the DeepseekV2 Network:** The total FLOPs of the network are the aggregate of FLOPs from all $T_1$ standard DecoderLayers and $T_2$ MoE DecoderLayers. Substituting $T_1 = 1$ and $T_2 = 11$ into the layer-wise FLOPs and simplifying the summation, we obtain the **total FLOPs formula**:

$$\text{Total FLOPs} = 12 \cdot \left(8nd^2 + 4n^2d\right) + 6nd\big[m + 11(km_1 + m_2)\big], \tag{25}$$

where the first term aggregates the self-attention FLOPs of all decoder layers ($T_1 + T_2 = 12$ layers in total), and the second term aggregates the FFN/MoE FFN FLOPs of the standard and MoE decoder layers, respectively.

## G. Limitations

One limitation of our work is that the proposed method is specifically designed for DeepSeek-OCR, as the goal of this paper is to further advance visual–text compression based on DeepSeek-OCR. In addition, although our approach achieves strong performance, the reductions in GFLOPs and prefill time are less pronounced than those observed in VQA tasks, due to the unique characteristics of OCR tasks. Exploring more effective strategies for achieving higher inference efficiency while maintaining performance remains an important direction for future work.

