# OpenReview forum: "RTPrune: Reading-Twice Inspired Token Pruning for Efficient DeepSeek-OCR Inference"
_ICML.cc/2026/Conference — ICML 2026 regular_

### Official Review · Reviewer_Af34 · 2026-02-15

**Soundness:** 3
**Presentation:** 4
**Significance:** 2
**Originality:** 3
**Overall Recommendation:** 4
**Confidence:** 4

**Summary:**

The paper suggests a method for post-hoc visual token pruning on top of DeepSeekOCR.
DeepSeekOCR reduces the number of visual tokens from 4096 to 64 - 400 visual tokens (depending on DeepSeekOCR configuration) and they want to reduce this further without hurting performance too much.
They observe that the important vision token embeddings tend to have high L2 norm. Using this observation they design two stage pruning algorithm. In the first stage they select high norm embeddings as anchors. Then they do a soft matching between the remaining tokens to the tokens selected in the first stage. They do so by sinkhorn iteration. Embeddings that are too far away from the anchors are discarded by a threshold.
They also give a heuristic to dynamically choose the amount of anchors per image dynamically. By combining a text density measure and embedding and inter token similarity measure.
They empirically show that their method is superior to competing methods with similar average number of visual tokens on OmniDocBench and olmOCR-Bench

Key contributions as written by the authors in the paper:
• We introduce RTPrune, a plug-and-play visual token
pruning method in DeepSeek-OCR which mimics the
reading twice behavior of the LLM via a two-stage
pipeline: retaining high-norm tokens and then merging
the remaining ones via optimal transport.
• We propose a dynamic pruning strategy to enable a better efficiency–accuracy trade-off, which combines the
post-encoding inter-token similarity and the original
textual density of the image.
• We conduct extensive experiments on various OCR
benchmarks, demonstrating that RTPrune consistently
achieves state-of-the-art under both a fixed pruning
ratio and our dynamic pruning strategy

**Compliance With Llm Reviewing Policy:**

Affirmed.

**Key Questions For Authors:**

No major questions. I did not find any unclear points that would change my evaluation.

**Limitations:**

Impact statement: While the authors include an impact statement, it is mostly focused on the benefits of efficiency. It does not discuss the risks of secondary compression in high-stakes domains (e.g., legal or medical OCR), where even minor information loss during pruning could have significant consequences.

**Strengths And Weaknesses:**

Strengths:
1. Well written and easy to follow
2. Empirical results Evaluated on olmOCR-Bench and OmniDocBench with edit distance / tree edit distance.
3. It compares against DeepSeekOCR, FITPRUNE (AAAI’25), NUWA (ICLR’26), and DIVPRUNE (CVPR’25) under the same setup.
4. Includes component ablations (Tables 5 and 6) that isolate the contributions of the ratio of kept high norm tokens r and token pruning by norm  and matching with OT for token merging with other models.
5. x1.23 prefill speedup compared to baseline

Weaknesses:.
1. Although there are some ablations on different models it still appears quite specific to deepseekOCR.
2. Not enough discussion about failure modes. For example, What about visual registers? It was shown that there is a tendency in VLLMs to have visual registers. Visual registers are usually not informative but have very high norms and we need to verify the impact of visual registers on this method.

Overall, the paper is simple to follow and understand.
It is not a groundbreaking paper.
There is a clear problem.
They notice a pattern they can use to create a heuristic for the problem.
They use well established methods for matching (Sinkhorn OT).
Results are better than the competition.

My main concerns are:
1. the amount of interest in the community since the scope is very narrow and very tailored to deepseekOCR.
2. Visual registers feel very in contradiction to the importance by norm rule.

---

> ### Author Rebuttal · Authors · 2026-03-29
>
> **Thank for your valuable input and appreciation on our work**
> # 1. Generalizability (for weakness 1 & concern 1&2)
> With the growing emergence of end-to-end OCR models (e.g. DeepSeek-OCR2, LightOnOCR, GLM-OCR), we expect the phenomenon of text-rich tokens having larger norms to generalize beyond DeepSeek-OCR. To verify that our method is not narrowly tailored to DeepSeek-OCR, we further evaluate its generalizability on several recently released end-to-end OCR models with OmniDocBench.
>
> | Method        | Visual Tokens↓ | Prefill(ms)↓ | Decode(ms)↓ |  Text↓   | Formula↑  |  Table↑   | Overall↑  |
> | :------------ | :------------: | :------: | :-----: | :------: | :-------: | :-------: | :-------: |
> | DPSKOCR2[1]   |      1083      |   59.1   |  16.8   |   0.05   |   90.54   |   86.94   |   90.93   |
> | CDPruner      |      560       |   57.6   |  16.6   |   0.45   |   50.30   |   39.61   |   68.28   |
> | DivPrune      |      560       |   57.6   |  16.6   |   0.27   |   65.77   |   57.67   |   85.97   |
> | Fitprune      |      560       |   58.3   |  15.9   |   0.22   |   65.89   |   62.23   |   70.76   |
> | Ours          |      560       |   57.6   |  16.6   | **0.14** | **85.07** | **80.44** | **88.64** |
> | LightOnOCR[2] |      2137      |   30.4   |  16.4   |   0.16   |   87.00   |   83.22   |   84.91   |
> | CDPruner      |      1238      |   23.4   |  16.0   |   0.22   |   88.45   |   73.42   |   79.96   |
> | DivPrune      |      1238      |   23.4   |  16.0   |   0.29   |   70.17   |   62.89   |   67.95   |
> | Fitprune      |      1238      |   27.8   |  16.1   |   0.94   |   6.33    |   4.92    |   5.85    |
> | Ours          |      1238      |   23.4   |  16.0   | **0.18** | **85.67** | **81.26** | **83.00** |
> | GLM-OCR[3]    |      4512      |   26.3   |  11.0   |   0.10   |   86.05   |   13.92   |   63.20   |
> | CDPruner      |      2783      |   17.1   |  10.7   |   0.16   |   77.31   |   5.15    |   55.39   |
> | DivPrune      |      2783      |   17.1   |  10.7   |   0.57   |   41.92   |   1.82    |   28.95   |
> | Fitprune      |      2783      |   18.3   |  10.7   |   0.49   |   57.62   |   12.65   |   40.92   |
> | Ours          |      2783      |   17.1   |  10.7   | **0.15** | **81.74** | **12.72** | **59.98** |
>
> **Remark 1:** Some base models do not include a CLS token in their visual encoders. Therefore, for fairness and convenience, we only compare with pruning methods that do not rely on a CLS token.
>
> **Remark 2:** For GLM-OCR, we use the `transformers` to implement, which outputs tables as plain text rather than HTML or LaTeX. Since OmniDocBench requires one of the latter two formats for table evaluation, the corresponding score is artificially low. We'll continue to investigate this issue and make corrections to the evaluation in future revisions.
>
> Additional experiments show: 1) Other pruning methods exhibit large performance variations across different OCR models, whereas our method remains consistently stable, preserving over **95%** of the original model performance across all tested models. 2) Our method achieves strong results on these metrics, offering substantial inference acceleration with minimal performance degradation, which further demonstrates both the effectiveness and the generalizability of our method.
>
> [1] DeepSeek-OCR 2: Visual Causal Flow. Arxiv26
>
> [2] LightOnOCR: A 1B End-to-End Multilingual Vision-Language Model for State-of-the-Art OCR.  Arxiv26
>
> [3] GLM-OCR Technical Report. Arxiv26
>
> # 2. Impact statement and future work (for weakness 2)
>
> **Purpose**: To the best of our knowledge, **this is the first work to investigate visual token pruning for OCR models, especially DeepSeek-OCR. We also release our code. We believe our study can serve as a starting point for future research on OCR-oriented pruning, and may also inspire subsequent efforts on leveraging OCR for context compression**.
>
> **Potential risks in some scenarios**: We acknowledge that, while our method is designed to improve efficiency, compression may introduce non-negligible risks in high-stakes domains such as legal or medical OCR, where even minor information loss could lead to significant downstream consequences. We agree that this limitation deserves more explicit discussion, and we will revise the Impact Statement to better reflect these potential risks. We sincerely thank the reviewer for this important perspective.
>
> **Future work**: Our current method is a simple yet effective solution motivated by an initial exploration of OCR models and tasks. Moving forward, we plan to analyze the remaining failure cases, particularly on challenging scenarios such as multi-column documents and long tiny text, and further study how visual registers may be incorporated to improve performance. We also plan to investigate more conservative and reliability-aware pruning strategies for high-stakes scenarios, supported by stronger mechanisms for detecting and preserving critical information.

---

> > ### Author Rebuttal · Reviewer_Af34 · 2026-04-03
> >
> > You answered my question regarding the paper being specific to deepseekocr. However, i still do not fully understand how the high norm assumption and the register tokens with high norm can go together

---

> > > ### Author Response · Authors · 2026-04-03
> > >
> > > # Clarification on the high norm assumption and visual registers
> > >
> > > We sincerely apologize for the misunderstanding, as we had mistakenly interpreted the “visual registers” as the “CLS” token.
> > >
> > > In our analysis in Sec. 3.2, we **observe that high norm, high attention, and high informativeness are strongly correlated but we does not claim that they are strictly equivalent**. Thus, it is entirely possible that DPSKOCR also contains some visual register tokens that exhibit high norms despite carrying little or no direct semantic or layout information.
> > >
> > > Whereas, **our method is not built on the assumption that every high-norm token is informative. Instead, by norm-based token selection, OT-based token merging and dyanmic pruning ratio, we aim to avoid discarding any token that may still contribute useful information**, so that complementary cues can be preserved as much as possible, even if some retained tokens may ultimately prove less useful. Therefore, the possible existence of visual registers does not invalidate our norm-based criterion.
> > >
> > > In future work, we plan to further investigate how such register tokens affect final OCR performance and to explore how they can be explicitly identified. We believe this direction could make our current method both simpler and more efficient.
> > >
> > > We sincerely appreciate this insightful comment, and please feel free to let us know if there are any further questions or concerns.

---

### Official Review · Reviewer_yBns · 2026-03-05

**Soundness:** 3
**Presentation:** 3
**Significance:** 3
**Originality:** 3
**Overall Recommendation:** 4
**Confidence:** 4

**Summary:**

This paper presents RTPrune, a two-stage visual token pruning method specifically designed to optimize the inference efficiency of DeepSeek-OCR. By supporting dynamic pruning ratio adjustment, the proposed approach demonstrates significant potential for practical applications.

**Compliance With Llm Reviewing Policy:**

Affirmed.

**Final Justification:**

My concerns have been addressed for the most part, and I maintain my positive assessment of this paper.

**Key Questions For Authors:**

Regarding the pruning ratio, what is the basis for setting it as $\phi(1 - \rho)$? Does this formula work out-of-the-box, or does it require calibration to fit different model scales? The paper should specify if any transformation is needed to keep the pruning intensity stable.

**Limitations:**

Yes

**Strengths And Weaknesses:**

**Strengths**
1. This paper evaluates how OCR and VQA tasks differ, identifying why current methods fall short, and proposes a novel two-stage visual token pruning strategy.

2. The proposed dynamic pruning ratio is a practical addition that directly benefits real-world OCR deployment.

3. The paper is well-structured and easy to follow.

**Weaknesses**
1. The paper lacks a comprehensive discussion of existing visual pruning strategies specifically designed for OCR. If no such methods exist, the authors should explicitly state this; otherwise, these methods should be included as baselines for a fair comparison.

2. There is a noticeable inconsistency in the baselines presented in Table 2. For example, DivPrune is only compared on DeepSeek-OCR-Small and Large, while the Tiny and Base variants are omitted. Similar gaps are observed for other methods like SparseVLM and VisionZip, which undermines the fairness of the comparative analysis.

---

> ### Author Rebuttal · Authors · 2026-03-29
>
> **Thank for your valuable input and appreciation on our work**
>
> # 1. Compared to other OCR model pruning methods (for weakness 1&2)
>
> To the best of our knowledge, there are currently no visual token pruning methods specifically designed for OCR models, and **we are the first work to investigate visual token pruning for OCR models, especially DeepSeek-OCR**. In fact, this absence of OCR-specific baselines is exactly why we run extensive experiments on DeepSeek-OCR, aiming to provide a broad and meaningful comparison in this setting.
>
> As for the apparent inconsistency in Table 2 and Table 3, we would like to clarify that this design was adopted to balance experimental cost and fairness. Rather than exhaustively running every method on every model variant, we randomly group methods under each setting and ensure that all compared methods are evaluated the same number of times under comparable conditions. In this way, the comparison remains balanced, while avoiding excessive experimental overhead. We agree that this protocol was not sufficiently explained, and we will describe it more clearly in the revised version.
>
> # 2. More explanation of dynamic pruning ratio (for question 1)
>
> 1) **Design**: The motivation for using $r_{\text{dyn}}=\varphi(1-\rho)$ is to jointly capture two complementary factors in OCR images: $\varphi$ measures the average token similarity, which reflects the redundancy of visual content, while $\rho$ measures the text density, which reflects how much informative textual content is present. Intuitively, a higher $\varphi$ suggests that more redundant visual tokens can be removed, whereas a higher $\rho$ indicates that pruning should be more conservative to avoid disturbing informative text tokens. Their product therefore provides a simple and effective way to adapt the pruning ratio to both redundancy and text richness.
>
> 2) **Range**: In practice, this formulation works out-of-the-box and does not require model-specific redesign. The effective range of $r_{\text{dyn}}$ naturally varies across models and images because both $\varphi$ and $\rho$ are data-dependent statistics. For example, on OmniDocBench with DPSK-OCR-Large, $r_{\text{dyn}}$ is typically around 0.16, where both the average token similarity and the text density are about 0.2. For images with tiny and dense long text, the pruning ratio can be as low as 0.1, while for images with large white or redundant regions, it can increase to around 0.4, as also reflected in Table 6 of the main text.
>
> 3) **Usage**: We would also like to clarify that the purpose of $r_{\text{dyn}}$ is not to define a rigid fixed ratio, but to provide a stable adaptive rule for balancing efficiency and accuracy. In our implementation, no additional transformation is required beyond normalizing $\varphi$ and $\rho$ to the range $[0,1]$, which allows the pruning intensity to remain naturally stable across different model scales. We will clarify this design motivation and its practical operating range more explicitly in the revised paper.

---

> > ### Author Rebuttal · Reviewer_yBns · 2026-04-03
> >
> > I'd like to thank the authors for their rebuttal. My concerns have been addressed for the most part, and I maintain my positive assessment of this paper.

---

> > > ### Author Response · Authors · 2026-04-03
> > >
> > > Dear **Reviewer yBns**,
> > >
> > > Thank you for your thoughtful feedback and for recognizing our efforts in the discussion. We think it is an important future direction for **existing vision token pruning methods to support end-to-end OCR models**. We also believe that **leveraging OCR for context compression** is a promising approach, and it may be a key focus of our next step. We sincerely appreciate your constructive comments and valuable insights. Please feel free to let us know if you have any further questions.

---

### Official Review · Reviewer_8iDb · 2026-03-12

**Soundness:** 3
**Presentation:** 3
**Significance:** 2
**Originality:** 2
**Overall Recommendation:** 4
**Confidence:** 4

**Summary:**

This paper proposes RTPrune, a training-free, plug-and-play visual token pruning method tailored to DeepSeek-OCR. The key idea is based on an empirical observation of a two-stage “reading twice” behavior during decoding: early layers attend to high-norm visual tokens, while later layers redistribute attention to remaining tokens. RTPrune first retains high-norm tokens and then merges the remaining tokens into the kept set via an optimal-transport assignment with a dustbin, and further introduces a dynamic, content-aware pruning ratio that combines inter-token feature similarity and image-level textual density. Experiments on OmniDocBench, olmOCR-Bench, and Ocean-OCR show improved efficiency accuracy trade-offs over prior pruning baselines, with up to 1.23× prefill speedup while retaining >99% accuracy on OmniDocBench.

**Compliance With Llm Reviewing Policy:**

Affirmed.

**Final Justification:**

The authors have addressed my concerns.

**Key Questions For Authors:**

I have some questions:

1. How is $r_{dyn}$ clipped in practice? What are the min/max pruning ratios?

2. Could author report a simple nearest-neighbor merge and/or a ToMe-style merge baseline to isolate the contribution of optimal transport?

3. Has the author evaluated RTPrune on particularly dense formula or table-heavy subsets with very small text?

4. What is the end-to-end runtime overhead of computing φ, ρ, the similarity matrix S, and running Sinkhorn (iterations, ε) relative to the prefill savings?

5. How sensitive are results to $\alpha$ and $z$?

6. I know the dynamic pruning ratio incorporates the text density of the Sobel operator, but Sobel performs poorly in edge detection for blurry/low-contrast documents. So how much will RTPrune's performance be affected in this scenario?

**Limitations:**

See weakness.

**Strengths And Weaknesses:**

1. This paper proposes a simple but effective two-stage strategy: norm-based selection plus optimal transport-based token merging with a dustbin to avoid harmful matches.

2. The dynamic pruning ratio $r_{dyn} = φ(1 − ρ)$, combining pairwise cosine similarity and sobel-based textual density, is a pragmatic content-aware rule for OCR.

3. Broad evaluation across three OCR-focused benchmarks and multiple DeepSeek-OCR variants, including both fixed and dynamic pruning regimes.

4. The core pipeline and intuition are clearly conveyed: two-stage reading --> two-stage pruning; figures and the algorithm box help operationalize the method.

Weaknesses:

1. Optimal transport with 100 Sinkhorn iterations and dense cosine similarity $O(N^2)$ adds non-trivial overhead, but the paper does not quantify the added compute vs net speedup from token reduction.

2. The sensitivity of  hyperparameter $\alpha$ is not reported.

3. When dense tables/math or low-contrast text dominate, L2-norm selection and Sobel density may be brittle.

4. No failure case analysis.

5. Some related works are not cited or compared, like Flowcut[1], HoloV [2], these are natural baselines for training-free token pruning.

[1] Flowcut: Rethinking redundancy via information flow for efficient vision-language models, NeurIPS 2025.

[2] Don’t just chase” highlighted tokens” in mllms: Revisiting visual holistic context retention, NeurIPS 2025.

---

> ### Author Rebuttal · Authors · 2026-03-29
>
> **Thank for your valuable input.**
> # 1. More details about $r_{\text{dyn}}$ (for question 1)
> In practice, $r_{\text{dyn}}$ varies across models and images. For DPSKOCR-Large on OmniDocBench, it is typically around 0.16, ranging from 0.1 for tiny dense text to 0.4 for images with large white areas. More details are provided in our second response to Reviewer yBns.
> # 2. More ablation studies on token merging (for question 2)
> We have conducted ablation studies on token merge in main text Table 5. Here, we further compare nearest-neighbor(NN) merge and ToMe-style merge. The results show that our method achieves the best overall performance, validating the effectiveness of OT-based feature matching for visual token propagation.
> | Method | Text↓ | Formula↑ | Table↑ | Overall↑ |
> | :-- | :--: | :--: | :--: | :--: |
> | Base + L2-norm | 0.18 | 79.00 | 80.18| 80.53 |
> | +NN | 0.18 |  79.51  | 80.26 | 80.69 |
> | +ToMe | 0.20 |  78.97  | 77.01 | 78.59 |
> | +OT(Ours) | 0.17 | 79.95 | 81.60 | **81.48** |
> # 3. Failure cases analysis (for weakness 3&4 & question 3&6)
> Our failure cases mainly stem from errors in spatial layout and text recognition. Spatial-layout failures are more common in formula-dense, table-heavy, and multi-column documents, while text-recognition failures are more evident for long tiny text (we copy part of main text table3 here for convenient lookup). We attribute these errors to two main factors:1) Unsuitable pruning strategies or pruning ratios may remove critical visual information, whereas our method(especially the dynamic pruning ratio based on spatial layout and textual information)helps mitigate this issue to some extent. 2) The performance of the base model is also crucial: **in general, the stronger the base model, the better the token-pruned model performs, and the smaller the results degradation**.
>
> Specially, sobel operator may fail on blurry or low-contrast documents, leading to a lower $\rho$ and thus a larger $r_{\text{dyn}}$, up to $\varphi$ in the worst case. Under this extreme setting, our method still retains around **96% of the original performance with a strong base model**, while the larger drop on Multi-Column and Long-Tiny-Text for weaker models further supports the second point above: **pruning performance is also strongly dependent on the capability of the base model.** In future work, we will further investigate how to improve performance under these challenging conditions.
> | Method | average $r_{\text{dyn}}$ | Multi Column↑ | Long Tiny Text↑ | Overall↑ |
> | :-- | :--: | :--: | :--: | :--: |
> | DPSKOCR-Base | - | 65.3 | 65.6 | 72.4 |
> | Ours | 16% | **45.1** | **34.8** | **63.7** |
> | Ours | 20%($\rho$=0) | 39.7 | 26.7 | 61.3 |
> | Ours | 25% | 31.8 |  20.4 | 57.7 |
> | Large | - | 68.2 | 76.9 | 75.5 |
> | Ours | 16% | **64.6** | **73.8** | **73.9** |
> | Ours |  20%($\rho$=0) | 64.0 | 71.7 | 72.8 |
> | Ours | 25% | 59.0 | 42.1 | 68.3 |
> # 4. Overhead of our work (for weakness 1 & question 4)
> Due to the uniqueness of OCR tasks, **$r_{\text{dyn}}$ is introduced to adaptively determine a better pruning ratio, and is orthogonal to each pruning method.** We believe this trade-off is worthwhile in our method: although it reduces prefill speedup by 23\%, it preserves nearly 99\% of the original performance.
>
> Regardless of $r_{dyn}$, vision-encoder(VE) pruning incurs higher overhead for same prefill speedup, while LLM-side pruning has lower overhead but more limited acceleration. Our method achieves a better trade-off between computational cost and prefill speedup.
> | pruning computation on DPSKOCR-Large | $\varphi$ | $\rho$ | $r_{dyn}$ | norm select | OT merge | Ours |  VE pruning | LLM pruning |
> | :-- | :--: | :--: | :--: | :--: | :--: | :--: | :--: | :--: |
> | runtime/prefill saving (ms) | 0.2/- | 5.3/- | 5.5/- | 0.1/18.2 | 4.5/18.2 | 4.6/18.2 | 6.7/18.2 | 1.1/3.4 |
> # 5. Sensitivity analyses of hyperparameters (for weakness 2 & question 5)
> **We have conducted ablation studies on the three hyperparameters $z$, $\alpha$, and $\tau$ in Appendix Sec. F**. The results show that our method is robust to all three hyperparameters, with moderate settings achieving the best balance between pruning efficiency and recognition accuracy.
> # 6. Compare with more works (for weakness 5)
> We have compared with 8 representative methods in the main text. Here, we further conduct additional experiments under a average 16% token reduction with both methods and will add citations, experimental comparisons, and discussion in the revised version.
> | Method | Text↓ | Formula↑ | Table↑ | Overall↑ |
> | :-- | :--: | :--: | :--: | :--: |
> | DPSKOCR-Large | 0.09 |  81.70 | 83.87 | 85.55 |
> | FlowCut[1] | 0.20 |  70.63 | 70.33 | 73.82 |
> | HoloV[2] | 0.27 |  74.29 | 66.49 | 71.43 |
> | Ours | 0.10 |  82.70  | 82.11 | **85.10** |
>
> [1] Flowcut: Rethinking redundancy via information flow for efficient vision-language models, NeurIPS25.
>
> [2] Don’t just chase” highlighted tokens” in mllms: Revisiting visual holistic context retention, NeurIPS25.

---

> > ### Author Rebuttal · Reviewer_8iDb · 2026-04-04
> >
> > Thank the authors' detailed rebuttal. my quesitons have been addressed, I will raise my score.

---

> > > ### Author Response · Authors · 2026-04-04
> > >
> > > Dear **Reviewer 8iDb**:
> > >
> > > Thank you for your kind follow-up. We are glad that our rebuttal helped address your questions, and we sincerely appreciate your time and consideration in raising the score.
> > >
> > > Please feel free to let us know if there are any further questions or points we can clarify during the discussion.

---

### Official Review · Reviewer_wSE8 · 2026-03-13

**Soundness:** 2
**Presentation:** 2
**Significance:** 2
**Originality:** 3
**Overall Recommendation:** 4
**Confidence:** 3

**Summary:**

RTPrune is a training-free token pruning method for DeepSeek-OCR.
The authors observe that the LLM decoder attends to high-norm visual tokens first in shallow layers, then shifts to remaining tokens in deeper layers.
Based on this, RTPrune selects high-norm tokens, merges pruned ones back via optimal transport, and adjusts the pruning ratio dynamically.  Everything happens once before the LLM.

**Compliance With Llm Reviewing Policy:**

Affirmed.

**Final Justification:**

The rebuttal addressed my three original concerns (lack of simple vision-side baselines, insufficient ablations on core design choices, and limited script coverage). The additional generalizability experiments across DeepSeek-OCR2, LightOnOCR, and GLM-OCR further demonstrate that the method is not narrowly model-specific. I adjust my overall recommendation from 3 to 4.

While the two-stage reading observation is an interesting finding, the resulting method builds on well-known techniques (norm-based selection, Sinkhorn OT, Sobel filtering), and the contribution is primarily empirical. I look forward to hearing other reviewers' perspectives on this point during discussion.

**Key Questions For Authors:**

- How does RTPrune compare to simple vision-side baselines (average pooling, stride downsampling) at the same token count? These work at the same stage and would be a fairer reference point.
- What happens when you replace L2 norm with other metrics like variance or entropy for token selection?
- Any results on scripts beyond English and Chinese?

minor comment:
- Several figures have very small text that is hard to read at normal size.

**Limitations:**

Yes, the authors note the method is DeepSeek-OCR-specific and gains are modest.

**Strengths And Weaknesses:**

Strengths:
- The pilot study (Table 1, Figure 2) clearly shows existing pruning methods fail on DeepSeek-OCR, even worse than random. This motivates the work well.
- Training-free and works across all DeepSeek-OCR modes without modification.
- Dynamic pruning ratio based on token similarity and textual density is a practical idea with clear gains.

Weaknesses:
- While vision-encoder pruning methods (DivPrune, CDPruner, VisionZip) are included, all of them rely on pruning criteria (diversity, textual relevance, encoder attention) that the paper itself argues are ill-suited for OCR. It would be useful to also compare against simpler training-free vision-side baselines at the same stage, such as average pooling or stride downsampling, to help isolate whether the gains come from the norm-based selection and optimal transport merging, or simply from operating at the vision-encoder stage with any reasonable strategy.
- Ablations mostly cover hyperparameter sensitivity (z, alpha, tau). Core design choices are not tested: L2 norm vs other importance metrics (variance, entropy), cumulative effect of each component (norm selection, then merging, then dynamic ratio), and the two factors in the dynamic ratio tested separately.
- All evaluation is on English and Chinese. The Sobel-based density estimation may not generalize to scripts with different stroke patterns.

---

> ### Author Rebuttal · Authors · 2026-03-29
>
> **Thank for your valuable input.**
>
> # 1. Compare with simple vision-side baselines (for weakness 1 & question 1)
> We further compare our method with simple vision-side baselines, including average pooling (realized by `AdaptiveAvgPool2d`) and downsampling (realized by `downsampling interpolation`), under a 25% token reduction setting. The results show that naive vision-side compression leads to substantial performance degradation and is insufficient for OCR models, whereas our method maintains much stronger performance and achieves the best overall result.
>
> | Method         | Text↓ | Formula↑ | Table↑ |  Overall↑ |
> | :-- | :--: | :--: | :--: | :-- |
> | DPSKOCR-Large | 0.09 |  81.70 | 83.87 | 85.55 |
> | average pooling | 0.67 | 39.59 | 34.38 | 45.52 |
> | downsampling | 0.45 | 62.37 | 61.28 | 59.55 |
> | Ours | 0.13 |  81.71  | 81.79 | **83.60** |
>
> # 2. More ablation studies on core design choices (for weakness 2 & question 2)
> We have presented ablations on token merging and dynamic pruning strategy in main text Sec. 4.4. Here, under an average 16% token reduction on DPSKOCR-Base, we further validate the cumulative effect of each component to validate our core design choices. 1) **Selection metric.** L2-norm performs better than other importance metrics. Variance-based selection also yields strong results, since features with larger norms often exhibit a wider value distribution and thus higher variance, but may miss tokens whose norm increases mainly due to an overall feature shift rather than larger dispersion. 2) **Merging method.** Combined with main text Table5, the results show that OT-based method consistently achieves the best overall performance under different selection criteria, validating the effectiveness of OT-based feature matching for visual token propagation. 3) **Dynamic pruning ratio.** We define the dynamic pruning ratio as $r_{\text{dyn}} = \varphi(1 - \rho)$, where both $\varphi$ and $\rho$ are in $[0,1]$. Since the two factors are multiplicatively coupled, we fix one while varying the other to better isolate its effect. The results further validate the effectiveness of dynamic pruning ratio based on the input. We will add these ablation studies in revised version.
>
> | select metric | merge method | similarity $\varphi$ | textual information $\rho$ | Text↓ | Formula↑ | Table↑ |  Overall↑  |
> | :--: | :--: | :--: | :--: | :--: | :--: | :--: | :--: |
> | - | - | - | - | 0.12 |  81.90  | 84.58 | 84.83 |
> | variance |   -   | ✔ | ✔ | 0.20 | 76.40 | 78.33 | 78.21 |
> | entropy |   -   |  ✔ |  ✔ | 0.53 | 61.83 | 51.99 | 53.61 |
> | L2-norm(Ours) |   -   |  ✔ |  ✔ | 0.18 | 79.00 | 80.18 | 80.53   |
> | variance |  OT(Ours)   |  ✔ |  ✔ | 0.19 | 76.91 | 79.51 | 79.20 |
> | entropy |  OT(Ours)   |  ✔ |  ✔ | 0.53 | 63.40 | 54.09 | 54.96 |
> | L2-norm(Ours) |  OT(Ours)   |  ✔ |  ✔ | 0.17 | 79.95 | 81.60 | **81.48** |
> | L2-norm(Ours) |  OT (Ours)  |  ✔ | 0.2 | 0.18 | 79.68 | 80.98 | 80.82 |
> | L2-norm(Ours) |  OT(Ours)   | 0.2 |  ✔ | 0.17 | 78.98 | 80.84 | 81.08 |
> | L2-norm(Ours) |  OT(Ours)   | 0.2 | 0.2 | 0.18 | 79.66 | 80.90 | 80.92 |
>
> # 3. Experiments on KITAB-Bench (for weakness 3 & question 3)
> We have evaluated our method on 3 OCR benchmarks in the main text. Here, we further include results on KITAB-Bench [1], which is a comprehensive benchmark for Arabic OCR. Under the same dynamic pruning setting, we compare our method with prior approaches. As shown in the table below, our method achieves the best overall performance on this benchmark, delivering substantial inference acceleration with only minimal performance degradation. In particular, it attains the highest CHrF score while maintaining CER and WER competitive with the unpruned baseline, further demonstrating the effectiveness of our method in multilingual OCR scenarios.
>
> | Method | Character n-gram F-score (CHrF)↑ | Character Error Rate (CER)↓ | Word Error Rate (WER)↓ |
> | :-- | :--: | :--: | :--: |
> | DPSKOCR-Large | 29.89 | 11.15 | 11.76 |
> | CDPruner | 27.81 | 12.92 | 12.81 |
> | DivPrune | 30.99 |15.15 | 15.56 |
> | Ours | **31.07** | **11.42** | **12.10** |
>
> [1] KITAB-Bench: A Comprehensive Multi-Domain Benchmark for Arabic OCR and Document Understanding. ACL25
>
> # 4. Readability (for minor comment)
> Thank you for pointing this out. We agree that the text in several figures is too small to read clearly. In the revised version, we will increase the font size and adjust the figure layout to improve readability.
>
> # 5. Generalizability (for limitation)
> As more end-to-end OCR models have been released recently, we believe that text-rich tokens in these models may likewise assign larger norms during training. Therefore, we evaluate the generalizability of our method on several recently released end-to-end OCR models, including DeepSeek-OCR2, LightOnOCR and GLM-OCR. Due to space limitations, **the results are provided in our first response to Reviewer Af34, which further demonstrates both the effectiveness and the generalizability of our method**.

---

> > ### Author Rebuttal · Reviewer_wSE8 · 2026-04-04
> >
> > Thank you for the thorough rebuttal. The simple vision-side baselines, component-wise ablations, KITAB-Bench results, and the generalizability experiments across DeepSeek-OCR2, LightOnOCR, and GLM-OCR collectively address my original concerns. I adjust my overall recommendation accordingly.
> >
> > That said, I want to be transparent about a remaining reservation: while the two-stage reading observation is an interesting finding, the resulting method builds on well-known techniques (norm-based selection, Sinkhorn OT, Sobel filtering), and the contribution is primarily empirical. I look forward to hearing other reviewers' perspectives on this point during discussion.

---

> > > ### Author Response · Authors · 2026-04-04
> > >
> > > Dear **Reviewer wSE8**,
> > >
> > > Thank you for your thoughtful follow-up and for updating your overall recommendation. We are pleased that our additional experiments helped address your original concerns.
> > >
> > > We also appreciate your transparency regarding your remaining reservation about the empirical nature of our contribution. Our primary contribution lies in showing that, for end-to-end OCR models, especially DeepSeek-OCR, norm-based selection and Optimal-Transport-based token merging can be integrated in a **simple yet effective** manner, guided by the model-specific two-stage reading behavior that we identify. **We believe this study can serve as a starting point for future research on pruning in end-to-end OCR models and may also inspire subsequent efforts to leverage OCR for context compression**.
> > >
> > > We look forward to the discussion with the other reviewers and would be happy to provide any further clarification if helpful.

---

### Decision · Program_Chairs · 2026-04-30

**Decision:**

Accept (regular)

**Comment:**

This paper proposes RTPrune, a training-free, two-stage visual token pruning method for DeepSeek-OCR. It selects high-norm tokens, merges others via optimal transport, and employs a dynamic pruning ratio. Experiments show better efficiency-accuracy trade-offs than prior pruning methods.


The main strengths of this paper are:
- A well-motivated method with a simple but effective training-free solution (Reviewer wSE8, 8iDb, yBns)
- A practical idea with a clear gain using a dynamic pruning ratio based on token similarity and textual density (Reviewer wSE8, 8iDb, yBns).
- Comprehensive evaluation across multiple benchmarks and  DeepSeek-OCR variants (Reviewer 8iDb), and component ablation (Reviewer Af34)
- Superior performance and prefill speedup against SOTAs (Reviewer Af34)


The main weaknesses of this paper are:
- Lack of comparison against training-free vision-side baselines (like pooling) to better isolate the contribution (Reviewer wSE8), and natural baselines like Flowcut, HoloV (Reviewer 8iDb).
- Insufficient ablation on core design choices (e.g., L2 norm vs. other metrics, cumulative effect of each component and the two factors in the dynamic ratio, token merging) (Reviewer wSE8, 8iDb).
- Limited script coverage beyond English/Chinese (Reviewer wSE8)
- Unquantified computational overhead (Reviewer 8iDb).
- More evaluation on dense formula or table-heavy subsets with very small text, blurry/low-contrast documents and different hyperparameters (Reviewer 8iDb)
- Lack of discussion about failure modes and the potential impact of visual registers (Reviewer Af34)


During the rebuttal phase, this paper received 4 Weak Accept. Even some concerns like empirical contribution and impact of visual registers exist, the AC acknowledges the paper's clear motivation, the simple yet effective method, and solid results. In this case, the AC tends to accept. The authors are strongly encouraged to incorporate a more thorough analysis of the ablation and a discussion on limitations (e.g., visual registers, generalization) in the final version.